# Endemic Burkitt lymphoma avatar mouse models for exploring inter-patient tumor variation and testing targeted therapies

Priya Saikumar Lakshmi[1,*], Cliff I Oduor[2,*], Catherine S Forconi[1], Viriato M'Bana[1], Courtney Bly[1], Rachel M Gerstein[3], Juliana A Otieno[4], John M Ong'echa[5], Christian Münz[6], Micah A Luftig[7], Michael A Brehm[8], Jeffrey A Bailey[2], Ann M Moormann[1]

**Endemic Burkitt lymphoma (BL) is a childhood cancer in sub-Saharan Africa characterized by Epstein–Barr virus and malaria-associated aberrant B-cell activation and *MYC* chromosomal translocation. Survival rates hover at 50% after conventional chemotherapies; therefore, clinically relevant models are necessary to test additional therapies. Hence, we established five patient-derived BL tumor cell lines and corresponding NSG-BL avatar mouse models. Transcriptomics confirmed that our BL lines maintained fidelity from patient tumors to NSG-BL tumors. However, we found significant variation in tumor growth and survival among NSG-BL avatars and in Epstein–Barr virus protein expression patterns. We tested rituximab responsiveness and found one NSG-BL model exhibiting direct sensitivity, characterized by apoptotic gene expression counterbalanced by unfolded protein response and mTOR pro-survival pathways. In rituximab-unresponsive tumors, we observed an IFN-α signature confirmed by the expression of IRF7 and ISG15. Our results demonstrate significant inter-patient tumor variation and heterogeneity, and that contemporary patient-derived BL cell lines and NSG-BL avatars are feasible tools to guide new therapeutic strategies and improve outcomes for these children.**

## Introduction

Endemic Burkitt lymphoma (eBL) was first described over 60 yr ago in pediatric patients in Uganda (Burkitt, 1958) and was the first viral-associated human cancer with the discovery of Epstein–Barr virus

(EBV) (Epstein et al, 1964). It is an aggressive B-cell non-Hodgkin lymphoma (B-NHL) that is etiologically linked to *Plasmodium falciparum* malaria endemicity and EBV infection in young children (Moormann & Bailey, 2016). Epidemiological data from Africa show that the average annual incidence of eBL is 2.4 per 100,000 children (Parkin et al, 2008). Conventional multiagent chemotherapies are universally used to treat eBL patients (Moormann et al, 2014); however, the survival rate for children diagnosed with eBL in sub-Saharan Africa hovers around 50% (Buckle et al, 2016), with mortality attributed to tumor aggressiveness, toxicity of high-dose chemotherapy, and relapsed or refractory disease after controlling for cancer stage and access to care in limited-resource settings (Ozuah et al, 2020). Devising strategies to improve survival using targeted treatments (i.e., rituximab), personalized treatment regimens, and expanding therapeutic options for relapsed or refractory eBL tumors would move the field of global oncology forward. Mutational profiles stratified by eBL tumor localization, survival outcome, and EBV content have been studied in depth (Kaymaz et al, 2017; Oduor et al, 2017; Panea et al, 2019). However, preclinical model systems are needed to understand clinical variation between patients, tumor heterogeneity, the integral role of EBV, and their combined implications for predicting individualized responses to current and novel treatment strategies.

Burkitt lymphoma (BL) diagnosed outside of Africa is ~30% EBV-associated and has a 10-fold lower incidence, in contrast to eBL tumors that are >90% EBV-associated (Hummel et al, 2006; Hämmerl et al, 2019). All BL tumors are diagnosed by aberrant *MYC* oncogene expression arising in the vast majority of cases from chromosomal translocations (t8;14, t2;8, or t8;22). These solid B-cell tumors appear to be relatively homogenous in their gene expression profiles, but have varied mutations associated with key

[1]Division of Infectious Diseases and Immunology, Department of Medicine, University of Massachusetts Chan Medical School, Worcester, MA, USA [2]Department of Pathology and Laboratory Medicine, Warren Alpert Medical School, Brown University, Providence, RI, USA [3]Department of Microbiology and Physiological Systems, University of Massachusetts Chan Medical School, Worcester, MA, USA [4]Jaramogi Oginga Odinga Teaching and Referral Hospital, Ministry of Medical Services, Kisumu, Kenya [5]Center for Global Health Research, Kenya Medical Research Institute, Kisumu, Kenya [6]Department of Viral Immunobiology, Institute of Experimental Immunology, University of Zürich, Zurich, Switzerland [7]Department of Molecular Genetics and Microbiology, Duke University School of Medicine, Durham, NC, USA [8]Program in Molecular Medicine and the Diabetes Center of Excellence, University of Massachusetts Chan Medical School, Worcester, MA, USA

Correspondence: ann.moormann@umassmed.edu
*Priya Saikumar Lakshmi and Cliff I Oduor contributed equally to this work

oncogenes, driver mutations, and are more distinct based on the EBV status of the transformed B cells than on the geographic location of diagnosis (Kaymaz et al, 2017, 2020; Panea et al, 2019). Common genes that are mutated in BL tumors include *MYC*, *TCF3*, *ID3*, and *DDX3X*, whereas EBV-containing tumors have more mutations secondary to activation-induced cytidine deaminase (Love et al, 2012; Abate et al, 2015). In addition, two divergent types of EBV are found circulating in African populations, yet EBV type 1 is more common in eBL tumors with only one-third containing EBV type 2 (Young et al, 1987; Kaymaz et al, 2020). EBV-associated tumors outside of Africa are predominantly EBV type 1, consistent with the predominance of type 1 virus circulating in the general population (Zimber et al, 1986). Therefore, models that include EBV variants are needed to dissect how EBV contributes to BL pathogenesis and impacts responses to different treatments.

EBV has growth-transforming potential in vitro, as observed in EBV-infected resting B cells that turn into lymphoblastoid cell lines and display latency III, characterized by the expression of six EBV-encoded nuclear antigens (EBV nuclear antigen 1 [EBNA1], EBNA2, EBNA3A, EBNA3B, EBNA3C, and EBNALP), and latent membrane proteins (LMP1 and LMP2). However, EBV-associated tumors generally display more restricted latency programs as an immune evasion strategy (Shannon-Lowe & Rickinson, 2019), with eBL tumors being classically categorized as EBV latency I, only expressing EBNA1, EBV-encoded small RNAs (EBERs), and BART miRNAs (Rowe et al, 1986). BL cell lines established over 50 yr ago, such as Daudi (Klein et al, 1968), Jijoye (Kohn et al, 1967), Namalwa and Raji (Pulvertaft, 1965), and more recently Akata (Takada et al, 1991) and Mutu (Gregory et al, 1990), established the dogma that EBV in BL tumors is latency I (Rowe et al, 1987). However, more in-depth studies found 15% of BL cell lines displayed Wp-restricted latency where EBNA2 is deleted, but the W promoter is active enabling the expression of EBNA1, EBNA3A, EBNA3B, EBNA3C, and BHRF1 (Kelly et al, 2013), and another study on eBL tumors in Malawian patients showed promiscuity in the latency pattern with the expression of lytic cycle genes (Labrecque et al, 1999; Xue et al, 2002). Altogether, these studies provide evidence that EBV protein expression in eBL tumors is more variable and dynamic than fully appreciated. EBV has been shown to provide a survival advantage by limiting apoptosis induced by the overexpression of *MYC*, and functional analysis of apoptosis-related proteins demonstrates that EBV suppresses apoptosis in multiple latency types including eBL (Price et al, 2017; Fitzsimmons et al, 2018). Current chemotherapies routinely disregard the presence of EBV in BL tumors. However, recent studies have focused on testing virus-targeting therapies such as small molecule inhibitors of EBV proteins, BH3 mimetics, EBV signaling pathway inhibitors, lytic activation–inducing drugs, and EBV-specific T cell–based immunotherapy (Kanakry & Ambinder, 2013). Emerging molecular characterization to dissect the role of EBV in BL pathogenesis provides a strong rationale for establishing contemporary patient-derived tumor model systems for interrogating the efficacy of virus-targeted therapies.

One of the most promising new therapies for CD20-positive lymphomas is the chimeric anti-CD20 monoclonal antibody rituximab that was approved by the US FDA in 1997 (Pierpont et al, 2018). In combination with conventional chemotherapy, rituximab has improved the 3-yr survival rate from 82% to 94% for children diagnosed with mature B-NHL in the USA (Minard-Colin et al, 2020). In the first rituximab clinical trial of African adults diagnosed with diffuse large B-cell lymphoma (DLBCL), the 2-yr survival rate was 55% when treated with rituximab and chemotherapy (Kimani et al, 2021), in contrast to chemotherapy only showing a 38% overall survival rate (Painschab et al, 2019). Rituximab has a myriad of ways to instigate tumor cell lysis including direct programmed cell death, complement-mediated cytotoxicity (CDC), antibody-dependent cellular cytotoxicity (ADCC), and antibody-dependent cellular phagocytosis (ADCP) (Maloney et al, 2002). The relative contributions of each mode of action are under debate and could vary between tumors (Boross & Leusen, 2012). Despite such exciting advances in targeted cancer treatments, they have been slow to reach patients in limited-resource countries.

Preclinical in vivo models, such as patient-derived xenografts (PDX) or patient-derived cell lines (PDC) implanted into immunodeficient mouse strains such as SCID or NOD-*scid IL2rg*^null (NSG) mice, are being used to explore cancer treatment efficacy, dosing, and toxicity (Walsh et al, 2017). These personalized mouse models have been named "avatars" because they provide a more physiological-relevant representation of the patient tumor for interpreting treatment responses. NSG mice engrafted with NHL tumors demonstrated responsiveness to combined anti-CD47 antibodies and rituximab mediated by ADCP (Chao et al, 2010), which was then tested in patients with refractory NHL (failure to respond to conventional chemotherapy) resulting in a 36% complete response, suggesting the addition of rituximab as a promising second-line strategy (Advani et al, 2018). Although humanized NSG mice have been used to study EBV oncogenesis, including EBV type 2 strains (Li et al, 2020), these tumors display DLBCL-like or Hodgkin-like lymphomas and were not linked back to evaluating the clinical presentation or outcomes of the patients.

To advance our understanding of inter-patient variation, we established five new patient-derived BL cell lines, compared them with long-established "old" BL cell lines, and then leveraged them to create PDC NSG-BL avatar mouse models. We then investigated inherent variation between the NSG-BL avatars and differences in response to rituximab. In parallel, we conducted comparative transcriptomic studies to further characterize tumors from pediatric Kenyan eBL patients, corresponding BL cell lines, and BL tumors grown in NSG mouse models.

## Results

### Establishment and characterization of patient-derived BL tumor cell lines

To generate new BL cell lines, we optimized previous protocols (Young et al, 1987) with our own novel modifications. An overall schema describes the steps leading to the establishment of BL cell lines, and the avatar model and list of evaluation assays (Fig 1A and detailed in the Materials and Methods section). In this study, we present five new patient-derived tumor cell lines (BL717, BL719, BL720, BL725, and BL740) established from fine-needle aspirates (FNAs) collected at the bedside from pediatric patients diagnosed with eBL in Kenya. Most of our cell lines grew as typical uniformly

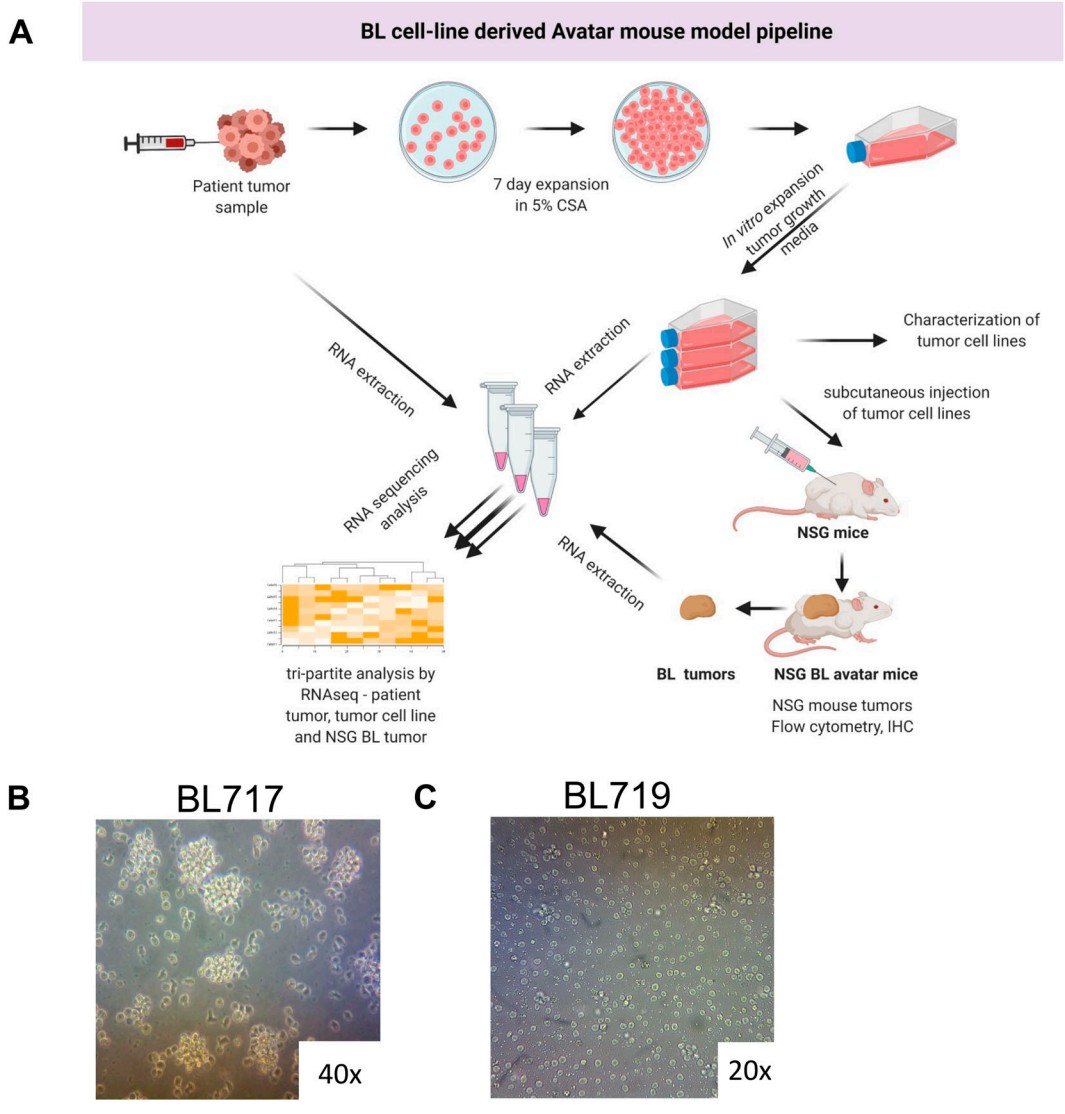

**Figure 1. Workflow for establishment and analysis of patient-derived BL tumor cell lines and NSG-BL avatar mice.**
**(A)** Key experimental steps summarizing establishment of BL tumor cell lines from fine-needle aspirates including in vitro culture adaptation and expansion, and transplantation into NSG mice. Transcriptomic analyses were performed on original eBL patient tumor samples, new BL cell lines, and NSG-BL mouse tumors. NSG-BL tumors were harvested and stained with antibodies for flow cytometry and formalin-fixed for immunohistochemistry staining. **(B, C)** Representative brightfield images of typical in vitro clustering of BL cell lines (BL717 shown, but also seen for BL720, BL725, and BL740); whereas (C) one tumor (BL719) grew as non-aggregating cells.

round B cells that formed aggregates in suspension (Fig 1B), but one (BL719) grew as a more dispersed single-cell suspension (Fig 1C). There were also differences observed in growth rates, with the EBV type 1 lines having doubling times ranging from 25 to 34 h and EBV type 2 lines doubling every 47–50 h (Table 1). The homotypic aggregation pattern and rate of replication of most of our new BL cell lines were similar to long-established standard BL cell lines, such as Raji, Mutu, and Akata. However, our findings demonstrate that some patient tumors require more time to establish and that the formation of clusters should not be used to visually determine whether a BL tumor cell line has been successfully established.

To molecularly confirm that our new BL tumor cell lines maintained key features of eBL patient tumors, they were characterized based on the expression of B-cell lineage markers (CD20, CD10, and CD19) and for the presence of *IGH* or *IGL/MYC* translocations and EBV. All our BL cell lines had uniform diagnostic features such as CD20 and CD10 expression, although there were minor variations in the abundance of surface marker expression, as determined by flow cytometry (Fig 2A). BL717 and BL725 cell lines had the greatest CD20 expression (>95% of the cells) followed by BL720, BL740 (~90%), and BL719 (~88%). All the BL cell lines were CD10-positive, with variability in expression ranging from 34 to 97% (Fig 2B) and threefold to 18-fold in mean fluorescence intensity above negative control (Fig 2C), thus surpassing the diagnostic threshold of >20% (Quintanilla-Martinez, 2017). All five BL cell lines contained a *MYC* translocation involving the immunoglobulin loci, determined by FISH. Four had t(8;14) involving the immunoglobulin heavy chain gene (*IGH*),

**Table 1. Clinical and molecular characteristics for Kenyan BL cell lines.**

| Cell line | Sex | Age (yr) | Site of tumor | Patient outcome | Doubling time (hr) | MYC translocation | EBV type | EBV load (copies/cell) |
|---|---|---|---|---|---|---|---|---|
| BL717 | Female | 8 | Abdomen | Survivor | 25.7 | t(8;14) | Type 1 | 26 |
| BL719 | Male | 3 | Jaw | Survivor | 50.2 | t(8;14) | Type 2 | 21 |
| BL720 | Male | 9 | Jaw | Non-survivor | 34.3 | t(8;22) | Type 1 | 28 |
| BL725 | Male | 5 | Abdomen | Survivor | 47.1 | t(8;14) | Type 2 | 20 |
| BL740 | Female | 7 | Jaw | Survivor | 33.4 | t(8;14) | Type 1 | 25 |

and one (BL720) had the t(8;22) translocation juxtaposing *MYC*/lambda light chain (*IGL*) (Fig 2D).

To detect EBV, we performed RNA in situ hybridization (RNA-ISH) for EBERs, the most abundant viral RNA in EBV-infected cells. We observed positive staining for EBERs in the nuclei of all five BL lines (Fig 2E). The frequency of EBER-positive cells was as follows: BL717 (73%), BL719 (84%), BL720 (46%), BL725 (86%), and BL740 (37%). In addition, we also measured EBV load by highly sensitive digital droplet PCR (ddPCR) and found a tight range of 20–28 EBV copies per cell across the new lines (Table 1). We also determined the EBV type using ddPCR with primers and probes targeting type-specific EBNA2 and EBNA3C, where deep-rooted variation exists (Sample et al, 1990). Three cell lines (BL717, BL720, and BL740) contain type 1 EBV, and the other two (BL719 and BL725) contain type 2 (Table 1). We found minimal differences in EBV gene expression between type 1 and type 2 tumors, consistent with our previous publication, using larger sample size (Kaymaz et al, 2017). No lines contained both types.

Our tumors had varied sites of presentation (jaw/abdomen), EBV type (type 1/type 2), and patient outcomes (survivor/non-survivor) post-chemotherapy (CHOP only) (Table 1), suggesting our ability to establish new BL cell lines was not skewed by these characteristics. Collectively, our findings demonstrate that contemporary BL cell lines retain eBL diagnostic criteria and importantly highlight inter-patient differences.

### Human gene expression signatures for eBL patient tumors and BL cell lines

We performed bulk RNA-seq on our new BL lines, the originating eBL patient tumors, and standard "old" BL cell lines to investigate their variance and concordance. To compare the overall transcriptomes, we performed principal component analysis (PCA) comparing four eBL patient tumor biopsies (BL740 patient specimen was not available for sequencing), the corresponding BL cell lines, and commercially available BL cell lines (i.e., Akata, Mutu, Namalwa, Raji, Jijoye, and Daudi) that were culture-adapted decades ago (Fig 3A). The PCA revealed distinct human gene expression signatures for each group, notably with the newly established BL cell lines clustering closer to the eBL patient tumors compared with older BL cell lines. Genes accounting for variation in principal component 1 (top loading values) were primarily involved in ion channel activity contributing to cell mitotic biochemical signaling, cell cycle progression, and proliferation (Table S1). This

implies that the long-established BL cell lines have likely acquired more robust growth activity and metabolic processes compared with freshly isolated patient tumors and newly derived BL cell lines. Genes associated with PC2 (top loading values) were involved in biological processes linked to immune system response and regulation, and the extracellular matrix (Table S1). This is consistent with differences because of the presence of leukocytes and stromal cells in the patient tumor samples and explains why we do not observe more subtle differences between individual patients in this type of bulk analysis.

We evaluated fidelity in expression profiles of key genes in our new BL cell lines compared with corresponding tumor biopsies (Fig 3B). Consistent with the protein expression by flow cytometry shown in Fig 2B, our new cell lines all expressed key genes used to diagnose BL, including *MYC*, *MS4A1*(CD20), *MME*(CD10), *MKI67*(KI-67), and *BCL6*. Differential gene expression (DGE) analysis showed minimal gene expression differences (456 DE of 11,235 genes), and no significant difference in the expression of BL marker genes in the BL cell lines compared with the patient tumors (gray dots in Fig 3B and Table S2). We performed DGE analysis between the old and new BL lines and identified 860 differentially expressed (DE) genes (Table S3). Gene set enrichment analysis (GSEA) identified the ATF2, serine/threonine kinase 33 (STK33), E2F3, HOXA9, and MEK pathways to be significantly enriched (false discovery rate [FDR] q <0.2; Table S4). The old BL lines were enriched for an oncogenic gene set associated with ATF2 activation (ATF2_UP.V1_UP; NES = 1.6; Fig 3C). *ATF2* plays a role in regulating cell proliferation, survival, and DNA damage response (Vlahopoulos et al, 2008). The new BL cell lines, in contrast, were enriched in the STK33 oncogenic gene set (STK33_UP; NES = 2.5; Fig 3D). *STK33* plays a role in regulating tumor proliferation (Scholl et al, 2009) (Table S4). As further confirmation of our findings, we used publicly available data from DLBCL tumors (Morin et al, 2010), the most closely related lymphoma, and eBL tumor biopsies from Ugandan patients as controls. Our PCA revealed that our Kenyan eBL patient tumors clustered with Ugandan eBL patient tumors and not DLBCL (Fig S1).

### Rituximab-mediated cell death of BL cell lines in vitro

Rituximab's efficacy is well documented in B-NHL and BL in the developed world (Plosker & Figgitt, 2003), but the overall therapeutic benefit for pediatric patients in Africa has yet to be determined. Here, we tested the sensitivity of our new BL cell lines to the direct induction of cell death by rituximab in vitro, with and without

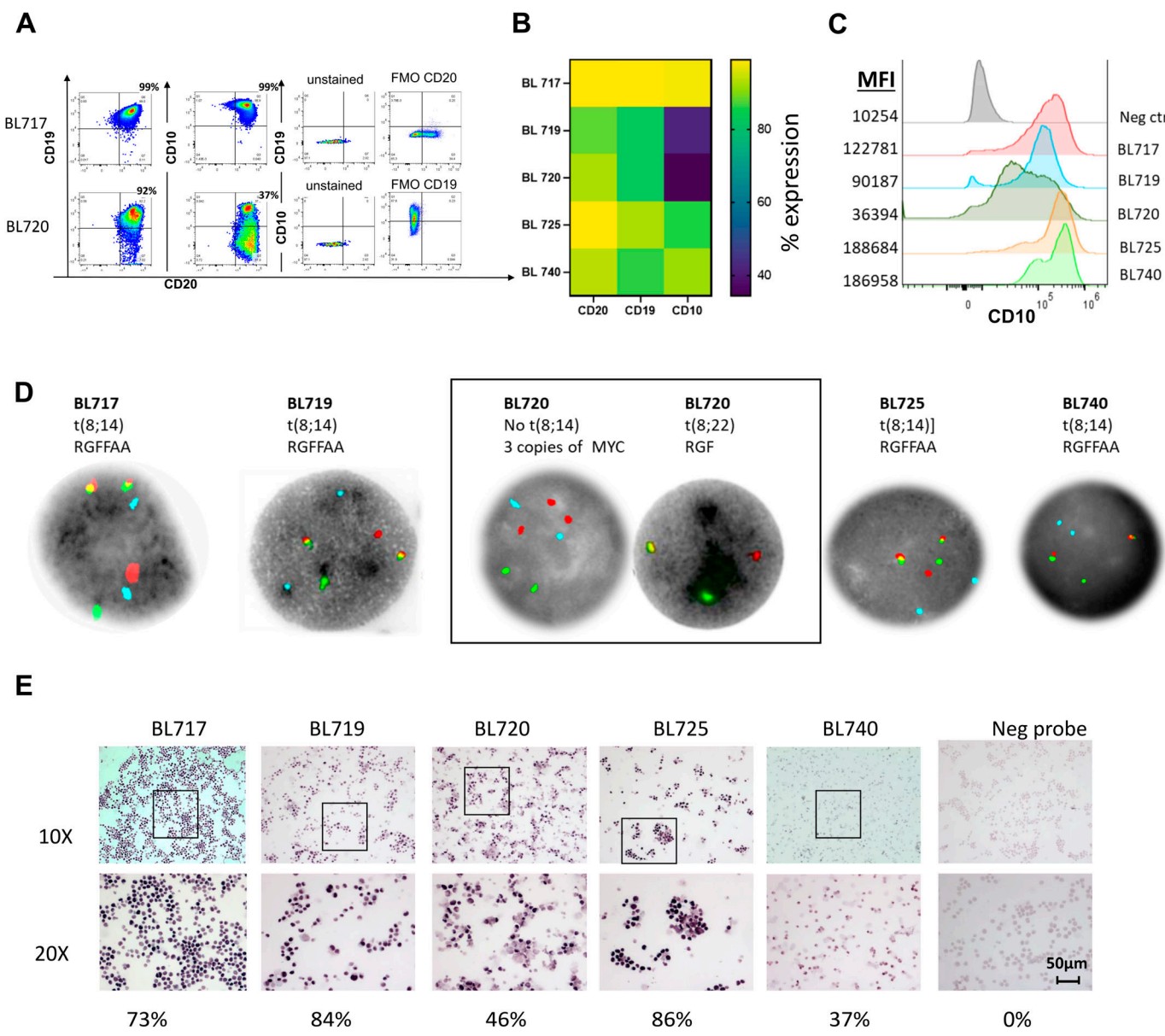

**Figure 2. Molecular phenotyping of patient-derived BL tumor cell lines.**
**(A)** Representative flow cytometry cytoplots showing surface expression of key diagnostic markers for BL (CD19, CD20, and CD10). Negative controls—unstained and fluorescence-minus-one plots for gating. **(B)** Heatmap of the mean of three independent experiments showing the frequency of CD20, CD19, and CD10 expression for newly established BL cell lines (BL717, BL719, BL720, BL725, and BL740) assessed by flow cytometry with CD20 (SD = 4.68), CD19 (SD = 10.58), and CD10 (SD = 14.67). **(C)** Histograms showing mean fluorescence intensity of CD10 expression across five BL cell lines, unstained control in gray. **(D)** FISH analysis using a *IGH/MYC* fusion probe, and chromosome 14 centromere probe shows classical t(8;14) MYC translocation for four of our BL cell lines, except for BL720, which shows no evidence of the MYC/IGH t(8;14) fusion. Aqua signal (A) = centromere of chromosome 8; red signal (R) = MYC (8q24); green signal (G) = IGH(14q32.3); and yellow signal (F) = MYC/IGH fusion. For BL720, further testing was done using the MYC break-apart probe set supporting IGL/MYC t(8;22) fusion, as shown on the right BL720 image grouped in the outlined box. **(E)** EBER staining (shown in purple) by in situ hybridization (RNA-ISH) assay demonstrated EBV-positive status for all of our five patient-derived cell lines. Negative RNA probe shows no staining. Percentage positive is indicated in the bottom of the images. Representative staining is shown (scale bar = 50 μm).

complement. Rituximab-mediated direct killing was measured by detecting phosphatidylserine exposure via Annexin V and 7AAD staining (Fig 4A). Early apoptotic cells are Annexin V+/7AAD– (Vermes et al, 1995), whereas late apoptotic cells and necrotic cells are characterized by both Annexin V+ and 7AAD+ staining. We did not observe early apoptotic cells across our new BL cell lines (Fig S2A). However, we found that BL740 showed significant cell death

based on Annexin V+/7AAD+ staining (39 versus 18% media alone, *P* = 0.04), as a result of direct rituximab activity, whereas the other four new BL cell lines were not affected (Fig 4B). We also tested old cell lines Akata (19 versus 15% media alone) and Mutu (18 versus 10% media alone) and observed significant differences in cell death in both cases (*P* = 0.02 and *P* = 0.001, respectively). Furthermore, we tested rituximab with complement (i.e., CDC) and found that our

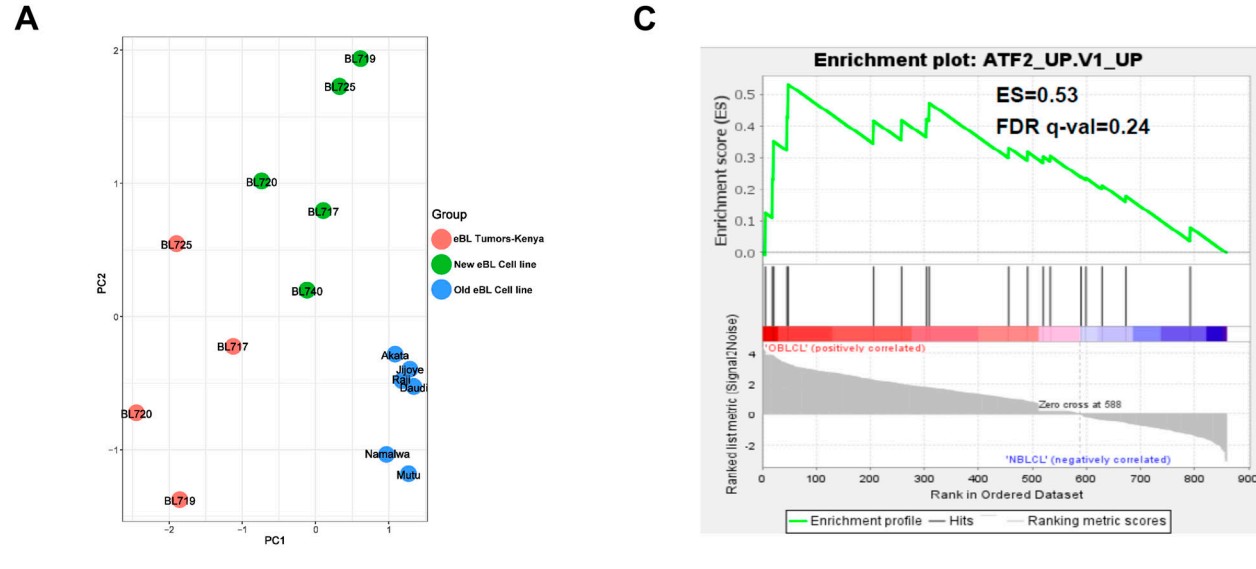

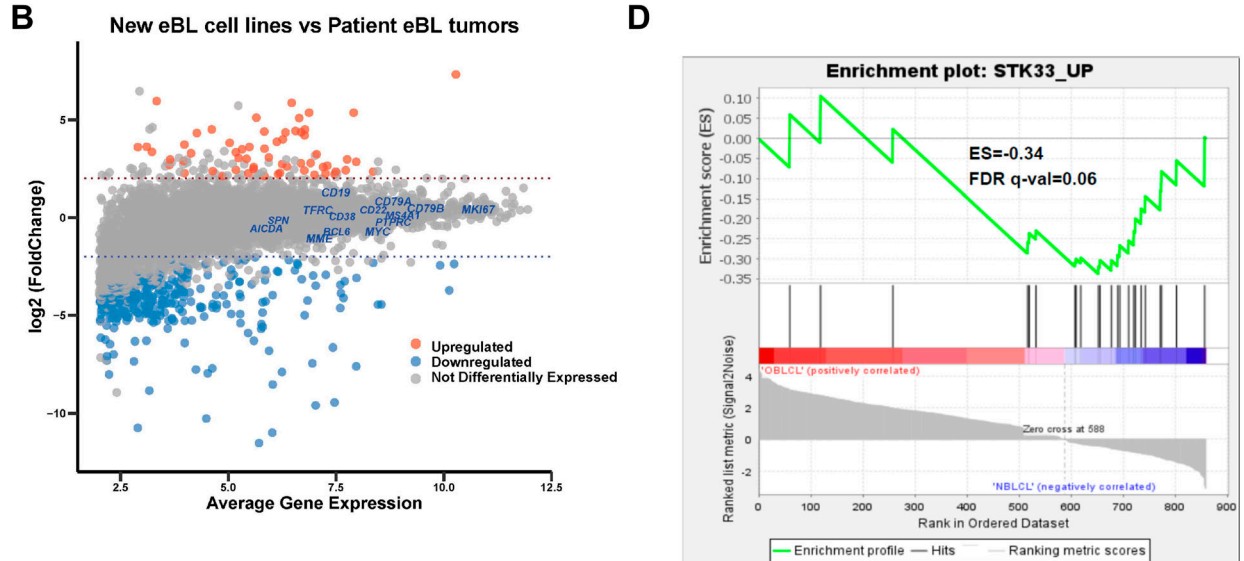

Figure 3.  **New BL cell lines maintain the gene expression profile of their parental eBL tumors but significantly differ from old BL cell lines.**
**(A)** Principal component analysis of gene expression comparing patients' eBL tumors (red dots), new patient-derived BL cell lines (green dots), and long-established "old" BL cell lines (blue dots, Akata, Mutu, Namalwa, Raji, Daudi, and Jijoye). **(B)** MA plot showing genes differentially expressed between the new BL cell lines and their corresponding patient eBL tumors. The MA plot illustrates a log₂FC (fold change) in the genes expressed against the average normalized expression of the genes expressed. Only 456 genes were identified to be differentially expressed of the 11,235 genes tested (Table S2). The red and blue dotted lines represent logFC cut-offs of 1.5 and −1.5, respectively. The red, blue, and gray circles represent up-regulated genes, down-regulated genes, and non-differentially expressed genes, respectively, in the new BL cell lines compared with the patient BL tumors. The curated gene names in blue font represent diagnostic BL markers, such as *MS4A1* (CD20); *MME* (CD10); *AICDA* (AID); and *MKI67* (Ki-67), which were not differentially expressed between the new cell lines and the patient tumors. **(C)** ATF2_UP.V1_UP gene set was enriched in old BL cell lines, in contrast to **(D)** STK33_UP gene set, which was significantly enriched in the new BL cell lines.

new BL cell lines showed no effect, with minimal cell death for NSG-BL740 tumors (Fig 4B; *P* = 0.1). Akata and Mutu displayed significantly increased cell death with complement (42 and 37%, respectively, both *P* = 0.001), suggesting rituximab action is likely to be complement-mediated in these cell lines, unlike our new BL cell lines. We also observed greater variation in rituximab responses using our new BL cell lines, likely reflecting tumor heterogeneity, which has been lost in old culture-adapted BL cell lines.

To further evaluate rituximab-induced cell death in selected BL cell lines, we measured intrinsic apoptosis determined by loss of mitochondrial membrane potential (MMP) (Ott et al, 2007). Cells with high MMP promote formation of red fluorescent JC-1 aggregates; thus, loss in membrane potential is measured as a decrease in red fluorescence (Fig 4C). Of our five new BL cell lines, only BL740 showed significant MMP loss compared with media alone, for both rituximab (50%, *P* = 0.004) and rituximab with complement (48%,

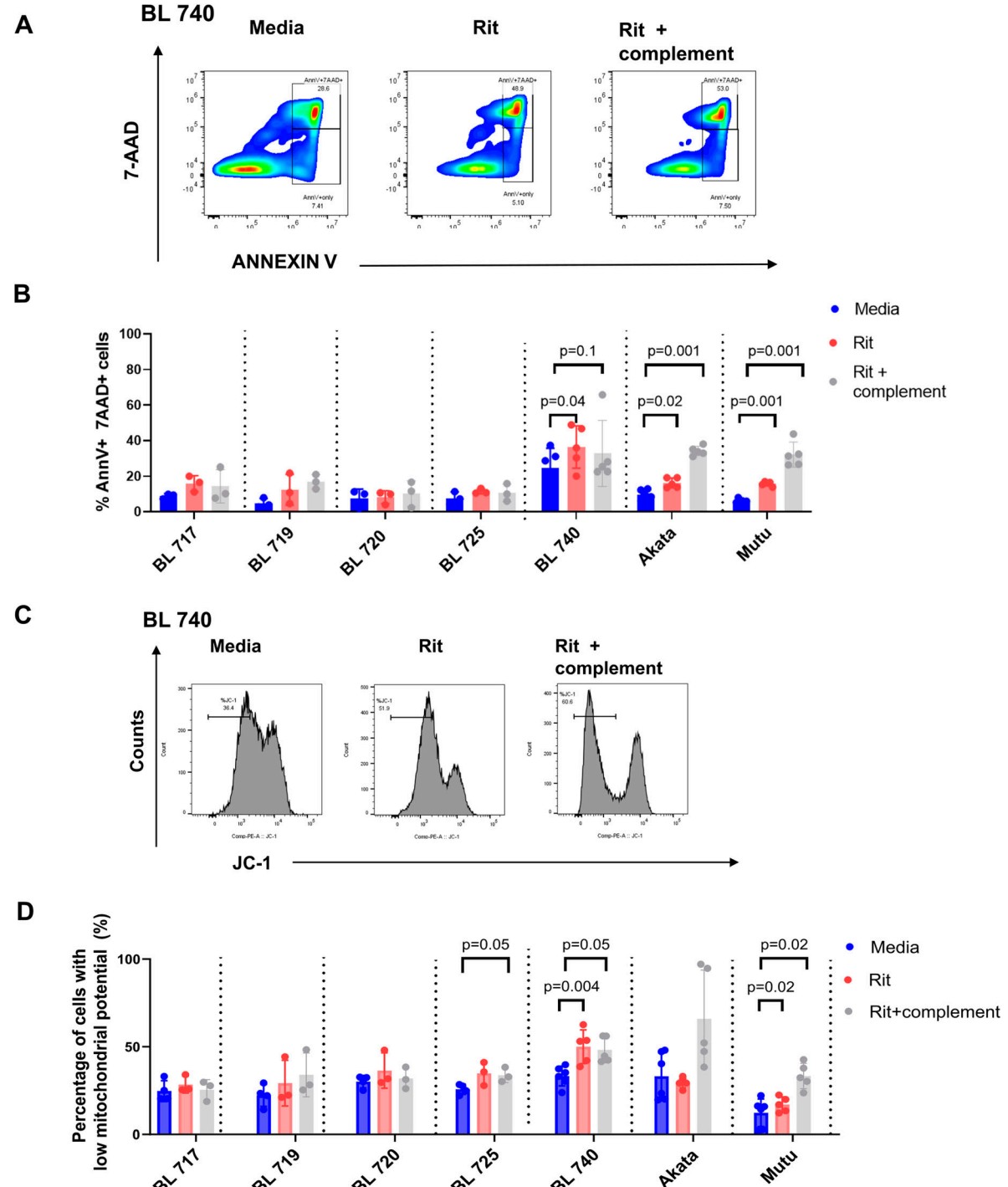

**Figure 4. Variation in rituximab-induced cell death in vitro across BL cell lines.**
**(A)** Representative cytoplots for Annexin V versus 7AAD staining in untreated and rituximab-treated (with or without complement) conditions. **(B)** Bar plot showing the percentage of Annexin V–positive and 7AAD-positive (% AnnV+/7AAD+) cells of our new BL tumor cell lines along with controls, Akata and Mutu, incubated with media alone (blue bars), Rit = rituximab (10 µg/ml) with 10% heat-inactivated human serum (red bars), or Rit+complement = rituximab with 10% complete human serum (gray bars). Data are represented as the mean, +/− SEM (three independent experiments). Statistical analysis was performed by parametric paired $t$ tests. $P$-values are shown when $P < 0.1$. **(C)** Representative histograms of JC-1 Red staining from the patient-derived BL740 cell line after being in culture with media only, rituximab (Rit), and Rit + complement = rituximab with 10% complete human serum (three independent experiments) by flow cytometry. **(D)** Bar plot showing the percentage of cells that had decreased mitochondrial transmembrane potential ($\psi$m) as determined by loss of red fluorescence of JC-1. Statistical analysis was performed by paired $t$ tests or Wilcoxon signed-rank tests, if they failed normality tests according to the Shapiro–Wilk normality test. $P$-values are shown when $P < 0.05$.

*P* = 0.05) (Fig 4D). BL725 also displayed MMP loss for rituximab with complement (*P* = 0.05). As for the old cell lines, Akata appeared to lose MMP with rituximab and complement (65%, *P* = 0.18) but this was not statistically significant, whereas Mutu had a minimal, yet significant, effect for rituximab without and with complement (17 and 33%, respectively, both *P* = 0.02). Taken together, these in vitro assays suggest that BL740 is partially sensitive to rituximab-mediated cell death and CDC, in the absence of cellular cytotoxicity (ADCC or ADCP).

### Characterization of BL cell lines in NSG-BL avatar mouse models

For each of our five BL cell lines, we established NSG-BL avatar mouse models (Fig 1A). In general, the NSG mice (92%, 22/25 mice) efficiently grew tumors. Interestingly, we found significant growth variation across BL lines despite injecting the same number of cells (Fig 5A). BL720 developed the fastest growing tumors (within 10 d), whereas BL719 generated tumors early (14 d). The other three cell lines slowly formed tumors after day 30. Relative proliferation rate of each BL-NSG tumor did not necessarily correlate with the corresponding BL cell line growth in vitro (Table 1). All NSG-BL avatars were followed up to 45 d with tumors harvested at this time point or predefined humane endpoint criteria, in terms of tumor burden (4,000 mm$^3$), necrosis of tumor, or loss in body weight (>20%). The BL720 avatars were the only ones euthanized early because of heavy tumor burden (4,000 mm$^3$, 3g). BL719 avatars showed the poorest survival but with low tumor weights and a low implantation rate (two of six mice) suggesting other pathologic mechanisms involved (Fig 5B). Poor survival was not due to an increased leukemic phase as they had minimal circulating tumor cells (0.5–1%) in the peripheral blood similar to other BL lines (Fig S4A). Also, there was no evidence of metastasis to the spleen or other organs for BL719, though we did observe metastasis for BL717 and BL720 avatars to the axillary lymph nodes (Table S5). We found that EBV type 2 tumors (BL719 and BL725) were smaller in size (BL719: 170 mm$^3$, 0.18g) compared with EBV type 1 tumors, with BL720 achieving the largest tumor sizes (3,000 mm$^3$, 3g) (Fig 3C).

### NSG-BL avatar tumors maintain their global transcriptome

We investigated the gene expression signature of NSG-BL avatar tumors to determine whether they had a similar profile compared with the original patient tumors. The correlation coefficient ($R^2$) was >0.9 showing that very few genes were divergent between the eBL patient tumors and NSG-BL tumors (Fig 5D). DGE analysis between the patient and avatar tumors only identified 486 genes that were DE, of the 9,580 genes identified, equating to only a 5% expression difference of total genes sequenced (Table S6). There was minimal difference in B cell–related gene expression arising from the NSG-BL tumor. Rather, differences were predominantly attributable to human immune cell signatures captured from the host tumor microenvironment including cytokine-mediated signaling, as seen in the gene enrichment map (Fig S3A and B). Thus, NSG-BL avatar tumors reflect the primary patient lymphomas and highlight inter-patient variation.

### EBV latency patterns in NSG-BL tumors

We analyzed EBV protein expression in our NSG-BL tumors by immunohistochemistry (IHC) and quantified them using QuPath software (Fig 5E). We detected EBNA1 expression in all our NSG-BL tumors but with varying frequencies ranging from 60% (BL717) to 10–20% (BL719 and BL720). Daudi (58%) and Mutu (24%) also displayed a similar range of EBNA1 positivity. We found that only BL719 and Mutu displayed the classic latency I pattern (EBNA1 only) associated with BL. In addition, we measured LMP1 (latency protein expressed in latency II and III) and EBNA2 (deleted in Wp-restricted latency but expressed in latency III). We found over 50% of BL740 cells expressing LMP1, suggesting a latency IIa pattern (Price & Luftig, 2015). Negligible (less than 2%) LMP1 expression for BL720 and BL725 was considered background noise as compared to Mutu, which is not known to express LMP1. Inclusion of EBNA2 expression for BL717, BL720, and BL725 suggests latency IIb (Price & Luftig, 2015); however, the expression of the other viral proteins for this classification is missing. Daudi cell lines have been previously reported to express EBNA3C in vitro (Jones et al, 1984). Taken together, our findings suggest that viral protein expression is not fixed as latency I for eBL tumors and may vary depending on immune pressure. Despite the atypical behavior of BL cell lines with respect to their EBV protein expression, we believe it will be important to monitor viral kinetics in NSG-BL tumors and how it affects responses to rituximab or other therapies.

In addition, we observed immediate–early protein BZLF1 expression in about 8–30% of cells within tumors (BL720, BL725, and BL740), which has the potential to convert EBV from latent to lytic cycle in productive versus abortive lytic cycles (Figs 5E and S6) (Ragoczy et al, 1998). Regulation of the Z promoter determines EBV latency, and more work is necessary to interrogate this phenomenon in the presence of immune pressure in mice. Our EBV positive control tumors Daudi and Mutu did not express BZLF1, suggesting that this may only happen in newly established BL cell lines.

### Rituximab sensitivity in NSG-BL avatars in vivo

We evaluated rituximab treatment across NSG-BL avatar models, by injecting BL717, BL720, BL740, and Daudi cell lines into the NSG mice to establish tumors, and then treating them with rituximab, as shown in the schema in Fig 6A. We increased our tumor cell concentration to 10 million cells to account for the slower growing tumors, except for BL720, which remained at 5 million cells per injection. Compared with the PBS group, NSG-BL tumors exhibited differences in direct rituximab sensitivity (Fig 6B), consistent with our in vitro studies shown in Fig 4. We observed reduction in tumor growth early for BL717, BL720, and Daudi, but this reduction was only persistent in BL740. At the end of the treatment (~27 d), we removed the tumors for further analysis and weights were assessed revealing a significant difference between treated and untreated BL740 mice (Fig 6C) compared with the other NSG-BL avatars. We performed TUNEL assay to detect DNA breaks that occur during the last phase of apoptosis and observed higher levels with rituximab-treated BL740 mice, compared with PBS-treated BL740 mice. There was no effect for the other NSG-BL avatars (Fig S2B and C). NSG-

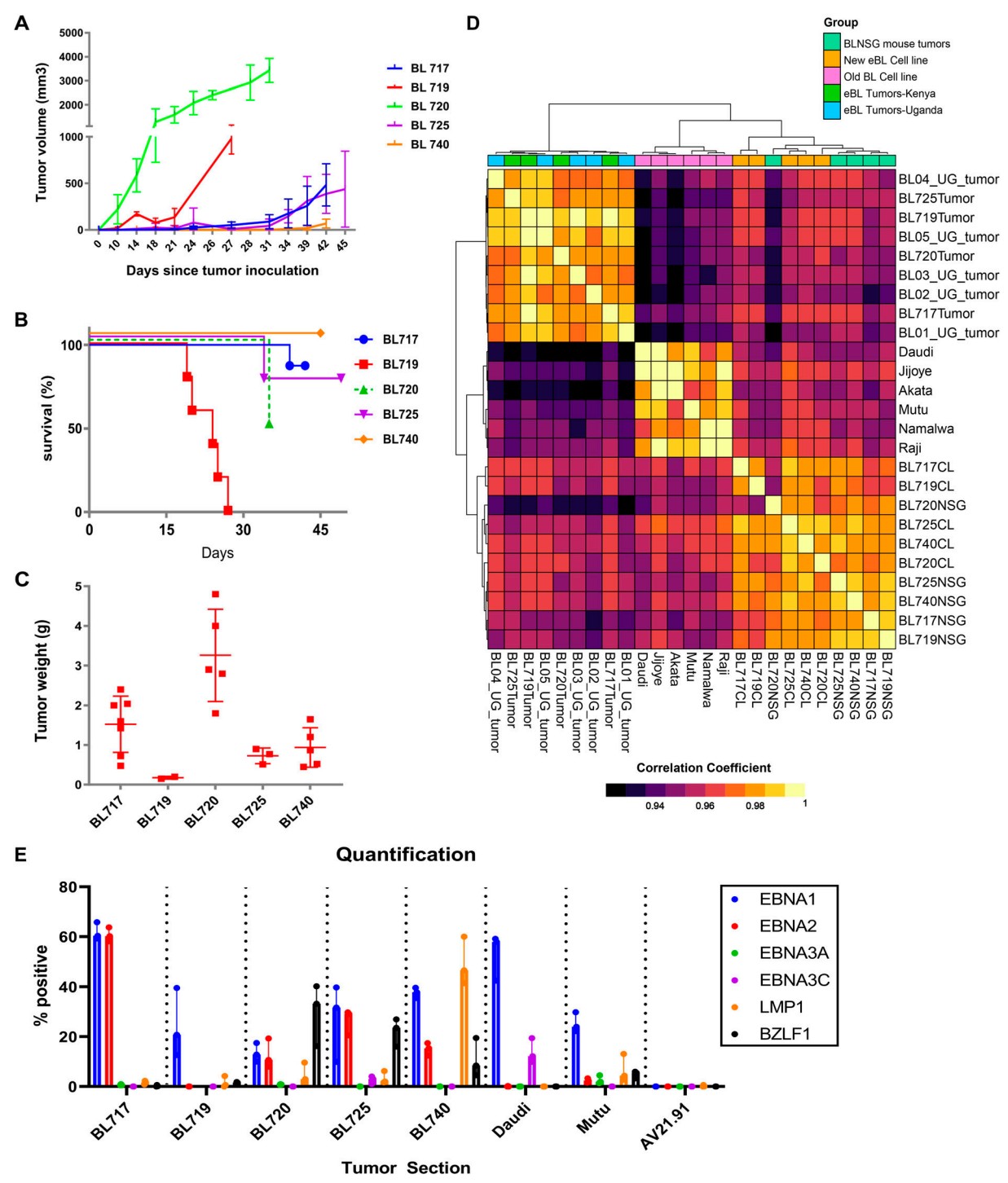

**Figure 5. Characterization of NSG-BL avatar mouse models: tumor growth kinetics, survival, transcriptome analysis, and EBV protein expression.**
Five patient-derived BL tumor lines injected into NSG mice varied in **(A)** tumor growth (N = 5 mice per group) and **(B)** survival kinetics (N = 4–8 mice per group). Dotted line for BL720 indicates that mice were euthanized because of reaching humane endpoints. The other NSG-BL tumors were followed up to at least 30 d. **(C)** Tumor weights were measured at experimental endpoints (N = 5–6 mice). *BL719 mice weights are from day 21, and other mouse tumors could not be recovered. BL720 had the highest tumor burden, and BL719 had the highest mortality. **(D)** Correlation heatmap of gene expression profiles from RNA sequencing of the eBL patient tumors (Tumor) from Kenya and Uganda, BL cell lines (CL) both old and new, and our NSG-BL tumors. The global gene expression patterns reflect biological relatedness among these groups, which showed high Pearson's correlation coefficient values >0.9, as demonstrated by the color scale illustrating the degree of correlation. Sample-to-sample clustering was based on the expression profiles of top 5,000 genes with highest correlation of variation (CV) values (calculated using regularized log-transformed expression data). The eBL patient tumor data from Uganda (UG) were sourced from a previous study (accession no.:PRJNA374464), where samples SRR5248346, SRR5248347, SRR5248348, SRR5248349, and SRR5248351 are represented as BL01_UG_tumor, BL02_UG_tumor, BL03_UG_tumor, BL04_UG_tumor, and BL05_UG_tumor, respectively. **(E)** IHC staining was performed for selected EBV proteins EBNA1, EBNA2, EBNA3A, EBNA3C, LMP1, and BZLF1 for the NSG-BL tumors along with Daudi and Mutu cell line–derived tumors and an EBV-negative control tumor (AV21.91). Akata cell lines failed to produce tumors in this mouse model. Quantification of tumor sections was performed on three fields per tumor using QuPath and is displayed on graph showing median values of the percentage area positive for each stain plotted.

Daudi avatars did not show any difference in tumor growth after rituximab treatment, which is consistent with previous mouse model studies, demonstrating the mechanism of killing is mediated by ADCC (Katano et al, 2020). As rituximab (hIgG) can also activate FcγR-bearing mouse cells for ADCP (Chao et al, 2010), we detected a greater frequency of mouse CD45+ cells in rituximab-treated BL740 tumors compared with untreated mice. The greater ratio of mCD45+ cells to tumor cells in BL740 tumors could be due to the smaller size of the tumors compared with other NSG-BL tumors (Fig S4B). Further analysis of the mCD45+ cell population showed no increase in Ly6G+ neutrophils or F4/80+ macrophages associated with rituximab treatment in NSG-BL740 avatars (Fig S4C). This implies that ADCP did not play a role in rituximab sensitivity for BL740. Rituximab treatment did not affect proliferation rate of the tumor cells, as measured by Ki-67 staining (Fig S4D). In summary, BL740 displayed sensitivity to rituximab in both in vitro cell lines and in vivo NSG-BL tumors in contrast to our other NSG-BL avatars.

### CD20 down-regulation with rituximab

One of the primary mechanisms of B-cell tumor resistance against rituximab is low CD20 expression (Pavlasova & Mraz, 2020), and many clinical studies have reported down-regulation of CD20 as limiting its therapeutic efficacy (Perz et al, 2002; Hiraga et al, 2009; Johnson et al, 2009). In our preclinical studies, we measured CD20 protein and gene expression (Fig 7A, top and bottom panels, respectively) in our NSG-BL tumors by flow cytometry and RNA-seq, respectively. All NSG-BL tumors treated with rituximab showed some reduction in CD20 expression compared with untreated mice. For BL717, BL740, and Daudi-NSG tumors, down-regulation of CD20 surface expression was particularly dramatic, whereas BL720 was barely changed. In contrast, lower *MS4A1* (CD20) gene expression was observed for BL720 and BL740 but not for BL717 or Daudi. We speculate that discordance in CD20 gene and protein expression is driven by transcriptional regulation (Pavlasova & Mraz, 2020), which could be exploited to improve rituximab efficiency.

### Rituximab-induced gene expression profiles in BL740 tumors

Of the three NSG-BL tumors treated with rituximab, only NSG-BL740 avatars experienced significantly reduced tumors (in the absence of complement, cell-mediated cytotoxicity). To further interrogate rituximab responsiveness pathways, we compared the expression profiles from rituximab and PBS-treated NSG-BL740 tumors. We identified 210 genes to be DE (logFC > 1.5 and FDR < 0.01) among the human transcriptome (Fig S5A and Table S7). Of the DE genes, 45 genes were up-regulated and 165 genes were down-regulated in the BL740 rituximab-treated tumors compared with the untreated tumors. Among the differentially expressed genes (DEGs), we observe an elevated expression of curated genes such as *PERP* (logFC = 3.3 and FDR = $2.77 \times 10^{-12}$), *CAV1* (logFC = 2.09 and FDR = $2.92 \times 10^{-9}$), *ACVR1* (logFC = 2.8 and FDR = $1.50 \times 10^{-13}$), *BCL2L11* (logFC = 1 and FDR = $1.39 \times 10^{-5}$), and *CASP3* (logFC = 0.6 and FDR = $1.5 \times 10^{-4}$) involved in apoptosis and regulating cell proliferation and down-regulation of *BCL-2* (logFC = −0.8 and FDR = $4.11 \times 10^{-6}$) (Fig 7B). To identify novel pathways induced by rituximab, given many more

genes were DE, we performed GSEA on the 210 DEGs, thus identifying six gene sets with significantly enriched representation (Table S8). Among the top enriched gene sets we observed were the unfolded protein response (UPR) pathway, mTORC1 signaling pathway, and apoptosis signaling in the rituximab-treated NSG-BL740 avatar (Fig S5B and Table S8). The enrichment of pro-survival–associated genes within the UPR and mTORC1 signaling suggests stress responses to mitigate apoptosis induced by rituximab. In addition, we stained our three NSG-BL tumors, as well as NSG-Daudi tumors for cleaved caspase-3 by IHC, and observed increased caspase-3 cleavage in rituximab-treated NSG-BL740 tumors, as well as NSG-BL720 tumors but not BL717 (Fig 7C). Because we only measured caspase-3 activity at the end of the treatment, we were unable to determine whether there was increased caspase activity earlier during the course of treatment.

### IFN-α pathway implicated in NSG-BL740 tumor responsiveness to rituximab

To gain more insights into the molecular mechanisms underlying differences between the rituximab-responsive NSG-BL740 and rituximab-unresponsive (NSG-BL717 and NSG-BL720) tumors, we performed differential expression analysis and identified 878 genes DE between NSG-BL717 compared with NSG-BL740 and 803 genes DE between NSG-BL720 compared with NSG-BL740 (Table S9). Of these DEGs, we observed up-regulation of B-cell survival genes such as *TNFRSF13C*, *IRF7*, and *STAT3* for NSG-BL717 and NSG-BL720 compared with NSG-BL740 tumors. GSEA performed on the normalized expression values of the DEGs identified the enrichment of 23 genes (such as *IFI44*, *IFI44L*, *EPSTI1*, *IL4R*) associated with the interferon-alpha (IFN-α) pathway in both NSG-BL717 and NSG-BL720 tumors compared with NSG-BL740 (Table S10). Apoptosis and the mTORC1 signal responses were inherently enriched in NSG-BL740 tumors compared with NSG-BL717 and NSG-BL720. By IHC, we confirmed key mediators in the type I IFN pathway, interferon-regulatory factor 7 (IRF7), and interferon-stimulated gene 15 (ISG15) expressed in NSG-BL717, NSG-BL720, and NSG-Daudi tumors but not NSG-BL740 (Fig 7D). NSG-Mutu tumors expressed ISG15 but not IRF7. Quantification of IRF7 and ISG15 expression is shown in Fig 7E. The role of the IFN-α pathway in eBL pathogenesis and rituximab resistance warrants further investigation.

## Discussion

It is estimated that 80% of cancer deaths occur in low- to middle-income countries, which has prompted numerous global oncology initiatives (Institute of Medicine, 2007). In this study, we provide a landmark road map from eBL patients in Kenya to state-of-the-art translational tools used to discover new cancer therapies. We optimized a new protocol for the efficient creation of patient-derived BL cell lines and demonstrated that our newly established BL cell lines retain high fidelity to the corresponding primary eBL patient tumors but differ in important oncogenic pathways from standard, long-established "old" BL cell lines. Of note, new BL cell lines displayed significant inter-patient variation in their growth kinetics, EBV latency, and rituximab sensitivity, which

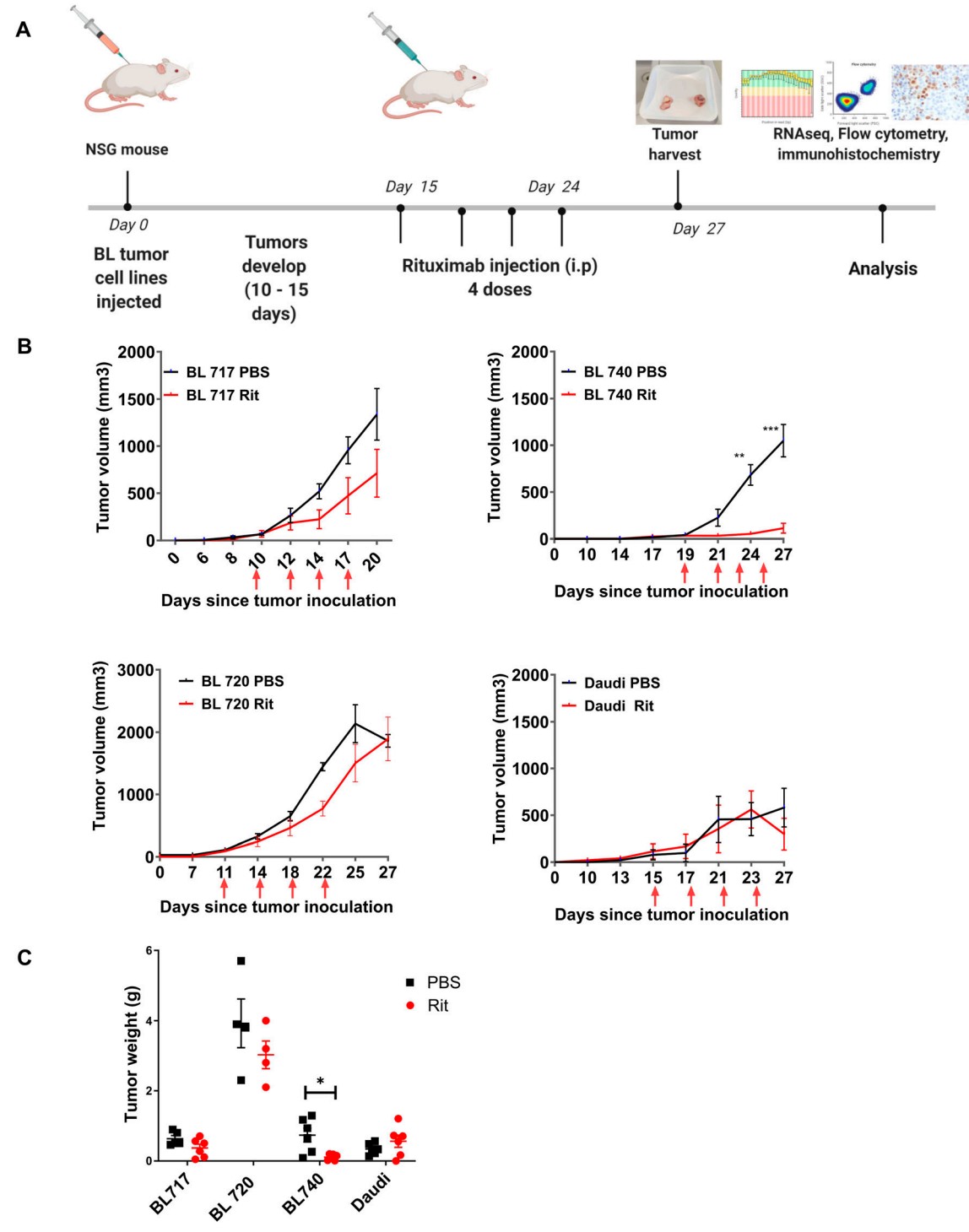

**Figure 6. NSG-BL avatar model supports a platform to screen targeted therapies.**
**(A)** Schematic showing experimental workflow starting with BL cell lines injected into NSG mice, tumors developing over 10–15 d, rituximab injections post-tumor development, and harvesting tumors for analysis. Tumors were analyzed using RNA-seq, flow cytometry, and IHC to gain insight into mechanisms involved in tumor regression or resistance to rituximab. **(B)** Tumor growth kinetics and **(C)** tumor weight for BL717, BL720, and BL740 lines injected into NSG mice treated with PBS (N = 6), black lines; or rituximab (N = 6), red lines, alongside the Daudi cell line as a control. Red arrows indicate time of each injection. BL740 tumors stopped growing in rituximab-treated mice and were significantly different from PBS mice on days 24 (**$P < 0.005$) and 27 (***$P < 0.0005$). Tumor weights collected at the end of the treatment also revealed significant differences in tumor weight (*$P < 0.05$). Statistical analysis was performed by an unpaired $t$ test.

underlie the call for individualized treatments. Using our new BL cell lines, we successfully established NSG-BL avatar mouse models that also demonstrate differential tumor kinetics, CD20 expression profiles, and responsiveness to rituximab in vivo, consistent with our in vitro studies. Combined with transcriptomic analyses, we were able to identify pathways, such as IFN-$\alpha$ and STK33, which are involved in oncogenesis in other tumors (Yang et al, 2016; Musella et al, 2017) but have never before been suggested as potential therapeutic targets for eBL patients. Thus, we demonstrate the feasibility of establishing contemporary NSG-BL model systems from African patients that have the potential to interrogate variability between patients. Our study is timely because rituximab, a standard of care combined with chemotherapy for BL treatment in high-income countries, is being introduced within pediatric eBL treatment regimens along with strengthening supportive care for oncology patients in low- to middle-income countries, including Kenya. Understanding rituximab failures within this context, including possible increased susceptibility to endemic infectious diseases, will be another important step toward improving overall outcomes for these children.

Using expandable patient-derived BL cell lines, we successfully established five different NSG-BL models and observed significant variation in tumor growth kinetics on the same mouse background. Most of our NSG-BL tumor-bearing mice exhibit sufficiently long survival for repeated dosing to simulate cancer treatment regimens and to test mechanisms leading to tumor regression or resistance. In the case of NSG-BL719, tumors grew early, and remained small but led to the rapid death of the mice. BL719 tumors exhibiting rapid softening and tumor necrosis suggesting these factors contributed to pathology similar to tumor lysis syndrome. Because BL is one of the most rapidly growing human tumors, our median time from patient biopsy to establishing cell lines ~3 wk, followed by ~ 4 wk to generate NSG-BL tumors, makes this pipeline feasible for studying clinically actionable markers.

Our NSG avatars include both EBV type 1 and type 2 tumors, which allowed us to further interrogate EBV latency. BL tumors endemic to Africa are classified as latency I, a more restricted pattern of antigen expression, expressing EBNA1 only (Rowe et al, 1986, 1987). However, there have been several accounts of other EBV antigens being expressed in BL tumors (Labrecque et al, 1999; Xue et al, 2002). Moreover, recent work by Gewurz and colleagues indicates that derepression of viral gene expression (both latent and lytic) can occur in the context of methionine restriction in both in vitro and in vivo xenografts of BL lines (Guo et al, 2022). Our study clearly demonstrates the expression of other EBV antigens such as EBNA2 and LMP1 in all but one of our NSG-BL tumors (Figs 5E and S6), suggesting they are not restricted, as described for classical BL cell lines such as Mutu and Akata. In fact, recent single-cell RNA-seq studies (SoRelle et al, 2021; Bristol et al, 2022) are now illuminating the extensive diversity of cell phenotypes in EBV–infected B cells and BL very likely will display a similar breadth of phenotypes when these tumors and cell lines are profiled. These studies collectively paint a more heterogeneous picture of the BL transcriptome than was previously appreciated. Characterizing and interrogating differences in EBV gene and protein expression between type 1 and type 2, and abortive versus productive lytic reactivation, are complex questions that will be addressed in subsequent studies.

We recognize that the biology of EBV latent infection is complex. The expression of EBNA2 (and other EBNAs) in the absence of LMPs has been described in early infection (Price et al, 2012; Price & Luftig, 2014, 2015; Messinger et al, 2019), EBV infection in humanized mice (McHugh et al, 2017), and EBV infection of chronic lymphocytic leukemia cells (Klein et al, 2013). The cyclic expression of LMP1 in latency III cells does not make the task of assigning EBV latency status any easier (Lee & Sugden, 2008; Brooks et al, 2009; Messinger et al, 2019). However, cellular gene expression associated with latency IIb versus (low LMP1) latency III indicates that it is feasible to dissect these states, especially when cell lines and xenografts do not seem to conform to strict latency types. Recent studies have shed light on the single-cell heterogeneity of the system, and future studies aim to characterize the clinical relevance of these latency states.

Rituximab, being highly successful in treating CD20$^+$ B-cell lymphomas, was our first choice of drug to test with our new BL cell lines and NSG-BL avatars. Rituximab induces apoptotic activity in vitro and has been previously studied in standard BL cell lines Raji, Daudi, and Ramos (Shan et al, 1998). Other in vitro studies using BL cell lines showed that cross-linking monomeric rituximab strengthened apoptotic signaling (Shan et al, 2000); however, studies in mouse models did not corroborate this finding (de Haij et al, 2010). In our study, one of the newly established BL cell lines, BL740, displayed direct cell death and complement-mediated cytotoxicity in vitro and direct cell death in vivo within NSG-BL740 avatars, consistent with reduced tumor size. The observed gene enrichment of pro-survival programs such as the UPR and mTOR signaling pathways implicates a cellular stress response, as well as enrichment of apoptotic signaling. BIM (*BCL2L11*) and PUMA (*BBC3*) have been identified as key pro-apoptotic genes repressed in latency I BL tumors (Fitzsimmons et al, 2018); however, we found increased BIM expression in BL740 that was up-regulated after rituximab treatment. This suggests alternate apoptotic mediators in BL740 tumors. Our NSG-Daudi avatars were not affected by rituximab; however, other studies have shown that Daudi engages CDC or ADCC when treated with rituximab (Li et al, 2009; Zhao et al, 2013). In addition, we observed enrichment of a type I IFN response (i.e., IRF7 and ISG15) in rituximab-unresponsive BL717 and BL720 tumors compared with the rituximab-responsive BL740 tumors. This suggests that type I IFN response signature may play a role in BL drug resistance, as has been observed in rheumatoid arthritis patients (Thurlings et al, 2010). It was curious to note that NSG-BL740 tumors that expressed the highest levels of LMP1 did not express IRF7, which was discovered in the context of EBV latency III infections and is a regulator of type I interferons (Zhang et al, 2001; Ning et al, 2003; Song et al, 2008). LMP1, an oncoprotein of EBV, activates IRF7 through a RIP- and TRAF6-dependent ubiquitination pathway. Inactive IRF7 present in the cytoplasm and viral infection triggers translocation into the nucleus to activate transcription. To further understand this, we measured ISG15, which is downstream of the IRF7 signaling pathway and found it was also not expressed in BL740. NSG-BL740 tumors also displayed BZLF1 expression, which has been shown to bind to IRF7 and repress its activity (Bentz et al, 2010) and also other regulatory mechanisms such as SUMOylation that could be regulating interferons in our model (Bentz et al, 2012). These exploratory findings require more detailed study of the LMP1/IRF7 pathway, if anti-viral or anti-cancer immunity is

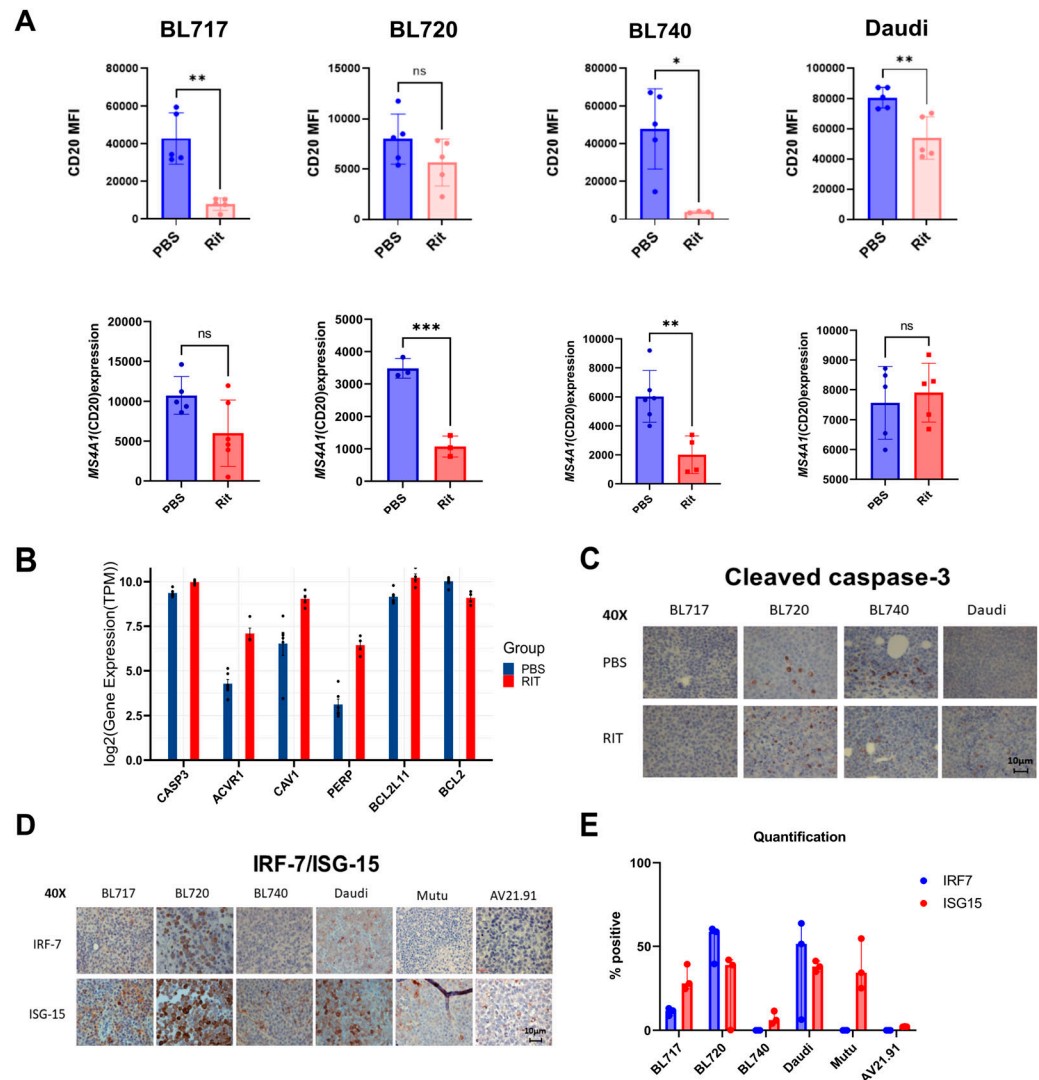

**Figure 7. Key biological pathways associated with rituximab responsiveness.**

**(A)** Rituximab treatment resulted in CD20 down-regulation for the NSG-BL tumors compared with untreated mice, at the level of both surface protein (by flow cytometry, top row) and transcripts (*MS4A1* expression by RNA-seq, bottom row). N = 3–5 mice per group; bars represent the mean ± SEM. CD20 surface antigen expression is measured by mean fluorescence intensity via flow cytometry, as shown in the top row (*$P < 0.05$; **$P < 0.005$; and ***$P < 0.0005$). **(B)** Significant gene expression differences in key apoptotic-associated genes were observed between NSG-BL740 rituximab-treated and untreated tumors. The blue bars represent the PBS-treated NSG-BL740 tumors, and the red bars represent the rituximab-treated NSG-BL740 tumors. Among these genes shown, there was an elevated expression: *PERP* (logFC = 3.3 and FDR = $2.77 \times 10^{-12}$), *CAV1* (logFC = 2.09 and FDR = $2.92 \times 10^{-9}$), *ACVR1* (logFC = 2.8 and FDR = $1.50 \times 10^{-13}$), *BCL2L11* (logFC = 1 and FDR = $1.39 \times 10^{-5}$), and *CASP3* (logFC = 0.6 and FDR = $1.5 \times 10^{-4}$), and there was down-regulation of *BCL-2* (logFC = −0.8 and FDR = $4.11 \times 10^{-6}$) in the RIT-treated compared with PBS-treated NSG-BL740 avatar mice. **(C)** IHC staining was performed on NSG-BL tumors (BL717, BL720, BL740, and Daudi) treated with PBS and rituximab against cleaved caspase-3 antibody. Representative images of staining shown at 40× magnification (scale bar = 10 μm). **(D)** IHC staining examined the expression of IFN-α–responsive proteins, IRF7 and ISG15, in NSG-BL tumors (BL717, BL720, and BL740) along with control BL tumors (Daudi and Mutu) and a non-BL, EBV-negative control tumor (AV21.91). Representative images of staining shown at 40× magnification (scale bar = 10 μm). **(E)** Quantification of IRF7 and ISG15 expression by IHC was performed on three fields per tumor using QuPath software and is displayed on graph with median values of the percentage area positive for each stain plotted.

mediated by ISG15 (Perng & Lenschow, 2018), and to determine whether type I IFNs are exerting a pro- or anti-tumor effect (Musella et al, 2017) in BL tumors.

One of the known caveats of rituximab is that some patients relapse because of the development of resistance (Hiraga et al, 2009). Lower CD20 expression levels have been shown to correlate with poor survival in DLBCL patients treated with rituximab (Johnson et al, 2009). However, there is alternate evidence suggesting that CD20 expression does not predict treatment effectiveness in B-cell lymphomas (Johnson et al, 2009). In our study, we observed down-regulation of CD20 expression in NSG-BL tumors, after repeated doses of rituximab. This means of rituximab resistance has been observed in Daudi, Ramos, and other CD20+ B-cell lymphomas (Jazirehi et al, 2007). Other mechanisms of CD20 down-regulation include CD20–antibody complex shaving by trogocytosis and antigenic modulation leading to internalization and

degradation of the CD20–antibody complex (Pavlasova & Mraz, 2020). Alternatively, refractory tumors could be due to the elimination of only the tumor cells expressing CD20 with the CD20-negative cells surviving, suggesting intratumor heterogeneity that should be monitored during the course of treatment. Strategies to sustain or increase CD20 surface expression include chromatin-modulating histone deacetylase inhibitors such as chidamide, FOXO1 inhibitors, and aurora kinase inhibitors (Pavlasova & Mraz, 2020). Future studies will test a panel of targeted therapies combined with rituximab in our NSG-BL avatars.

SCID mouse models have been used to explore other mechanisms of action engaged by rituximab, such as CDC that requires complement and ADCC involving natural killer (NK) cells. We chose the NSG-BL mouse model as a launching point for the next set of experiments using humanized NSG mice to interrogate NK (Herndler-Brandstetter et al, 2017) and T-cell interactions (Walsh et al, 2017) and human complement–sufficient NSG-Hc mice (Verma et al, 2017). PDX models have been used to validate ibrutinib (Bruton's tyrosine kinase inhibitor) resistance in B-cell lymphoma (mantle cell lymphoma) and identified regimens to overcome drug resistance (Zhang et al, 2017). However, when optimizing and translating such mouse models, we are cognizant of altered immune signatures within pediatric eBL patients. Our studies of Kenyan children have clearly demonstrated the prevalence of unconventional cytotoxic T cells (Falanga et al, 2017) and CD56$^{neg}$ NK cells (Forconi et al, 2018) that may have a profound impact on the efficacy of antibody-based cancer therapies.

Both our newly established BL cell lines and old BL lines exhibit a BL-like phenotype; however, cellular processes involved in oncogene-driven pathways, such as cell migration, proliferation, and survival, showed significant differences. We found ATF2 activation–induced genes relatively overexpressed in long-established BL cell lines, such as Namalwa and Ramos (Walczynski et al, 2014), putatively because of *MYC*-induced stress leading to apoptosis. It could be argued that ATF2 enrichment in culture-adapted cell lines is due to the lack of protein turnover in downstream processes, thus resisting apoptosis. We also observed that genes associated with STK33 and HOXA9 up-regulation were relatively enriched in new BL cell lines. STK33 phosphorylates pro-apoptotic BAD (*BCL-xL*) and suppresses apoptosis via mitochondrial pathways (Scholl et al, 2009), whereas HOXA9, together with MYC, is known to help maintain the expression of multiple anti-apoptotic pathways (Miyamoto et al, 2021) that could improve survival of the cell lines. STK33 may also bind directly to *MYC* and increase its transcription (Yang et al, 2016). Exploring oncogenic pathways in contemporary BL cell lines is an area of active investigation, and here we would like to propose classifying BL tumor subtypes based on their potential to respond to clinically actionable targets.

In summary, our NSG-BL avatar models highlight inter-patient variability and show promise for being a useful tool for identifying new therapeutic strategies that could improve outcomes for children diagnosed with eBL in resource-limited settings. We demonstrate that patient-derived tumors have the potential to improve our understanding of EBV (type 1 and type 2) biology and eBL pathogenesis.

# Materials and Methods

## Human study participants

Ethical approval was obtained from the Scientific and Ethics Review Unit (SERU) at the Kenya Medical Research Institute (KEMRI) and the Institutional Review Board at the University of Massachusetts Chan Medical School, Worcester, USA. Written informed consent was obtained from adults and from parents of minor study participants in accordance with the Declaration of Helsinki. Children diagnosed with cancer were enrolled at Jaramogi Oginga Odinga Teaching and Referral Hospital (JOOTRH) in Kisumu, Kenya. Two independent pathologists confirmed diagnosis by cytopathology and May–Grunwald–Giemsa staining. Patient BL tumor samples were further characterized by transcriptome and mutational profiling to confirm BL diagnosis (Kaymaz et al, 2017).

## Establishment of "new" patient-derived BL tumor cell lines

FNAs from patients presenting with solid tumors were collected at the bedside directly into 1X PBS, pH 7.4, with 10% HI-FBS (heat-inactivated FBS, Gibco). The FNA samples were gently passed through a 70-μm cell strainer, to dissociate any tumor clumps into a single-cell suspension. The tumor cells were washed once in RPMI media supplemented with 15% FBS, 1X Pen/Strep, and 5 μg/ml cyclosporine A (CSA), and spun at 300$g$ for 5 min. The tumor cell pellets were resuspended in 1 ml RPMI containing CSA that interferes with IL-2 synthesis and thereby restricts the growth of T cells that could be cytotoxic to the B-cell tumors. This cell suspension was then divided into four wells of a 48-well non-tissue culture–treated plate (Nunc 48-Well Plate, Round, Cat no.: 150787; Thermo Fisher Scientific), and 250 μl of RPMI+CSA media was added to each well for a total volume of 500 μl per well. Cells were incubated in 5% CO$_2$ at 37°C for at least 5 d without being disturbed. After 5 d, cell cultures were visually inspected using an inverted microscope at 100× magnification (Nikon TMS) and supplemented with fresh media by gently pipetting to disperse any cell clumps. Half the contents of each well (250 μl) was transferred to an adjacent well in the plate to expand the cultures. An additional 250 μl of tumor cell growth media (85% RPMI media supplemented with 10% heat-inactivated human serum AB [0.1 μm sterile-filtered; GemCell], 1% L-glutamine [200 mM, 100X; Gibco], 1% Pen/Strep [100X; Gibco], 1% non-essential amino acids [NEAA, 100X], 1% sodium pyruvate [100X], and 1% Hepes [1M; Gibco]) was added to each well. The tumor cells were allowed to grow undisturbed at 37°C and 5% CO$_2$ for another 7 d. Our patient-derived BL cell lines grew free in suspension as clumps of cells or as single cells that did not adhere to the culture plate (Fig 1). After 7 d, tumor cells were passaged and aliquots were cryopreserved every 3 d on reaching their optimal density of ~10 × 10$^6$ cells/ml. All the cell lines were free of contamination by mycoplasma, and these newly established eBL cell lines will be made available to the scientific community as a shared resource.

## Commercially available "long-established" BL cell lines

Standard BL-derived cell lines, established decades ago, Namalwa (Pulvertaft, 1964), Jijoye (Kohn et al, 1967), and Daudi (Klein et al,

1968), were obtained from ATCC (American Type Culture Collection). Mutu I cell lines (Gregory et al, 1990) were kindly provided by Jeff Cohen (NCI), whereas the Akata cell lines (Takada et al, 1991) were provided by Kenzo Takada (Hokkaido University). The cells were maintained in complete RPMI-1640 medium (Gibco) supplemented with 10% heat-inactivated FBS (Gibco) and antibiotics (100 U/ml penicillin and 100 μg/ml streptomycin) at 37°C in a humidified 5% $CO_2$ incubator following the established protocols.

### FISH staining

FISH analyses were performed at the Clinical Duke Cytogenetics Laboratory at Duke University Health Systems Clinical Laboratories and carried out on cell lines. The Vysis LSI IGH/MYC/CEP 8 Tri-Color Dual Fusion FISH Probe Kit (04N10-020; Abbott laboratories) was used to detect the t(8;14) (q24;q32) reciprocal translocation involving the IGH and MYC gene regions, and also a centromere enumeration probe (CEP 8, D8Z2) as a reference for the copy number of chromosome 8. For BL720, we also performed interphase FISH analysis using a break-apart probe to test for any *MYC* rearrangement (Ventura et al, 2006). The 8q24 *MYC* rearrangement was detected using the Vysis LSI MYC dual-color break-apart probe (05J91-001; Abbott Laboratories) for BL720 alone, because it failed to show translocation in t(8;14). This probe set detects rearrangement of the *MYC* locus at 8q24.2 without identification of the translocation partner.

### EBV quantification and typing

For each patient-derived BL cell line established, we isolated DNA from the first passage preserved in RNAlater using the Monarch Genomic DNA Purification Kit (NEB). We used ddPCR to determine EBV load by amplifying EBV BALF5 and human *β*-actin gene, using primers and probes as previously described (Forconi et al, 2020). This method does not require a standard curve because the technique gives absolute nucleic acid quantification based on positive droplet count numbers (Vogelstein & Kinzler, 1999). The duplex ddPCRs were prepared in a total volume of 20 μl, which contained 10 μl of ddPCR Supermix for probes (no UTP) (Bio-Rad Laboratories), and two sets of each primer and probe combination (0.9 μM of primers and 0.25 μM of probes). In this experiment, we used DNA extracted from the Namalwa cell line as our EBV positive control, and BL41 DNA as our EBV negative control. The Bio-Rad Automated Droplet Generator (AutoDG; Bio-Rad Laboratories) was used to ensure consistent droplet generation. After the ddPCR, the number of positive and negative droplets was counted by the QX200TM Droplet reader (Bio-Rad) and EBV viral loads were quantified as EBV copies/ng human DNA.

EBV type was determined by ddPCR using primers and probes targeting the EBNA2 and EBNA3C genes. Namalwa and Jijoye cell line DNAs were used as EBV-type positive controls for EBV type 1 and type 2, respectively. We targeted two EBV type–specific genes to confirm the EBV type and to eliminate incorrect identification in case we had a cell line with an intertypic virus as previously reported (Kaymaz et al, 2020). The EBV typing protocol was a duplex reaction for the EBNA2, with a set of primers and probes targeting EBV type 1 and another set of primers and probes targeting EBV type

2 EBNA2. The EBNA3C typing reaction on the contrary had a set primer designed to amplify a region in the EBNA3C gene for both EBV types 1 and 2, while using two distinct probes targeting EBV type 1 EBNA3C and EBV type 2 EBNA3C genes. The reaction volume and reaction temperature conditions were similar to those used in the viral load ddPCR protocol.

### EBER in situ hybridization

EBER in situ hybridization of cytospin-fixed cells was performed using the Novocastra Predilute EBER probe kit (Leica Biosystems) following the manufacturer's protocol. In brief, cells were stained with the fluorescein EBER probe (ISH5687-A) followed by the BCIP/NBT detection kit (NCL-ISH-D). The specimen shows positive staining indicated by dark blue/black stain. Each run had a positive and a RNA negative control probe (ISH5950-A). We defined a case as EBER-positive when most (>50%) of the tumor cells had moderate-to-strong signals.

### IHC

Briefly, mouse tumors were fixed for 24 h in 10% (vol/vol) neutral buffered formalin (Thermo Fisher Scientific), paraffin-embedded, and sectioned into 4-μm sections. They were deparaffinized in xylene and rehydrated serially in 100%, 95%, and 75% ethanol and water, respectively, and pretreated in a 10 mM citrate buffer, pH 6.0, for 20 min in a water bath (98°C) followed by cooling. Endogenous peroxidase was blocked in 0.3% hydrogen peroxide. Then, the sections were incubated with a 1:100 diluted primary antibody for 1 h at room temperature and washed, and the slide was incubated with an appropriate species-specific secondary antibody for 1 h at room temperature, followed by visualization using DAB (Liquid DAB+ Substrate/Chromogen System) according to the manufacturer's recommendation (Code K3468; Agilent Technologies), and counterstained with hematoxylin stain. To detect EBV protein expression in the mouse tumors, we stained for EBNA1 (supernatant 1: 100; Monoclonal Antibody Core Facility & Research Group), BZLF1 (clone BZ1, sc-53905; Santa Cruz Biotechnology), LMP1 (clone [cs1-4], ab78113; Abcam), EBNA2 (clone PE2, ab90543; Abcam), EBNA3A (ab16126; Abcam), and EBNA3C (ab16128; Abcam). We also stained for cleaved caspase-3 (clone 5A1E, mAb #9664; Cell Signaling Technology), IRF7 (clone G-8, sc-74472; Santa Cruz Biotechnology), and ISG15 (clone F-9, sc-166755; Santa Cruz Biotechnology). Images were captured using NIS-Elements software (BR 2.3; Nikon). Patient melanoma tumors were obtained from the UMass Chan Medical School Cancer Avatar Institute (IRB ID: H00004721) and passaged in NSG mice to deplete the human leukocytes present within the tumor microenvironment (Maykel et al, 2014). The patient-derived xenograft melanomas (PDX melanomas) were then processed into ~2 × 2 × 2 $mm^3$ pieces or a single-cell suspension, and either tumor fragments or cells (2.5 × $10^6$) were transplanted subcutaneously to the right flank of NSG mice (Yao et al, 2019).

### Flow cytometry

Cell surface expression of tumor markers was detected by flow cytometry. Cells were resuspended in PBS containing 2% FBS. The

cells were adjusted to $1 \times 10^6$ cells/ml and blocked using 5% normal human serum, before antibody staining. Cells were stained with antibodies for 30 min at 4°C and were fixed using BD Cytofix/Cytoperm (BD Biosciences) for surface staining, and intracellular staining was performed using the transcription factor buffer set (Cat no.: 562574; BD Biosciences) according to the manufacturer's recommendation, only in the case of Ki-67. Cells were analyzed on a four-laser Cytek Aurora (Cytek Biosciences) by acquiring at least 100,000 events, and data analysis was done using FlowJo v10.0 (BD Biosciences). We used Zombie near IR (BioLegend) to gate on live cells for flow cytometry except in the case of Annexin V staining where we use 7-aminoactinomycin D (7AAD), a vital dye to assess cell death, instead. Our gating strategy includes gating on forward scatter versus side scatter, followed by gating for singlets, selecting for live cells and then focusing on our markers of interest. We employed fluorescence minus one to determine our gates in dot plots. Fig S7 confirms our cell lines were exclusively B cells.

Antibodies and reagents: The following antibodies were used for staining tumor cell lines and NSG-BL tumors via flow cytometry from BioLegend: BV510 anti-CD20 (clone 2H7), PE/Cy7 anti-CD10 (clone HI10a), AF647 anti-CD19 (clone HIB19), PE/Cy5 anti-human CD45 (clone HI30), BV605 anti-mouse CD45 (clone 30-F11), BV785 anti-mouse Ly6G (clone 1A8), AF700 anti-mouse F4/80 (clone BM8), and BV421 anti-Ki-67 (clone Ki-67).

## Library preparation and RNA sequencing

Total RNA was extracted from the patient FNA samples, the resulting BL cell lines, and NSG-BL mouse tumors using the Monarch Total RNA extraction kit (New England Biolabs). Starting with 1–5 μg total RNA, we enriched mRNA molecules using oligo-dT magnetic beads to capture polyA-tailed mRNA molecules. We then proceeded to prepare strand-specific RNA-seq libraries following the NEBNext Ultra II Directional RNA Library Prep Kit for Illumina (NEB). Final library concentration and fragment size were confirmed with a Qubit 4 fluorometer (Thermo Fisher Scientific) and fragment analyzer (Agilent), respectively. These libraries were sequenced with paired-end reads (2 × 75 bp) on the NextSeq 550 system (Illumina, Inc.). Raw sequencing data files from this study are deposited in the NCBI's database of Genotypes and Phenotypes (dbGaP) with accession number phs001282.v2.p1.

## Gene expression analysis

DGE analysis was performed using standard methods. Briefly, sequenced reads were first checked for quality using FastQC (Andrews, 2021) and sorted based on the unique sample indexes identified by Novobarcode (Novocraft Technologies). Residual Illumina 3′-end adapter sequences introduced during the cDNA synthesis were trimmed using cutadapt (Martin, 2011). After residual adapter removal, the reads were mapped to a concatenated human–mouse genome to filter out mouse transcriptome contamination. Reads that did not map to the mouse sequence of the concatenated genome were kept for downstream analysis of human gene expression arising from the BL tumors. Mapping to the human–mouse concatenated genome was done to all samples to minimize the batch effect. The mouse filtered paired reads were then aligned to a human

transcriptome index built by RSEM (RNA-seq by expectation–maximization) (Li & Dewey, 2011) using Gencode annotation version 19 for protein-coding transcripts and hg19 genomic sequence. RSEM calculated strand-specific expected read counts for each gene, and gene expression count matrices for each sample were generated for downstream statistical analyses that were performed on R software (https://www.R-project.org/) (version 3.5.3). For each tumor type, we calculated correlation coefficients between primary BL tumor samples, cell lines, and DLBCL tumors using the 5,000 most variable genes, as these genes are the most likely to be biologically informative. To show that our BL tumors, newly derived BL cell lines, and the NSG-BL mouse models all have a BL-like gene expression profile, we compared our expression data with those of previously published BL tumors collected from Uganda (accession no.: PRJNA374464). We also downloaded previously published data from DLBCL patients (a germinal center–derived B-cell lymphoma that exhibits a similar gene expression profile to BL) (dbGaP accession no.: phs000532.v2.p1), to discern any similarities to other GC B-cell lymphomas, apart from BL.

DGE analysis between patient BL tumor samples, resulting BL cell lines, and NSG-BL tumors was performed using the R package edgeR (Robinson et al, 2010), which implements a trimmed mean of M-values normalization and a negative binomial approach (Robinson & Oshlack, 2010). We removed from the analysis all genes for which all counts per million values were lower than 5 counts per million and also filtered out RBC-associated genes such as *HBA1*, *HBA2*, *HBB*, and *HBD*. To control for multiple testing, we applied the Benjamini–Hochberg (BH) procedure with a threshold for statistical significance set at an adjusted $P < 0.01$ and FDR < 0.01. Batch effects and unwanted sources of variations related to different sequencing platforms were removed using sva (Leek et al, 2012; Leek, 2014).

## GSEA

Ranked GSEA was performed with software version 4.1.0. The GSEA algorithm calculates an enrichment score reflecting the degree of overrepresentation at the top or bottom of the ranked list of the genes included in a gene set in a ranked list of all genes present in the RNA-seq expression data. A positive enrichment score (ES) would indicate a gene set enrichment at the top of the ranked list; a negative ES indicates gene set enrichment at the bottom of the ranked list. Gene sets were considered significantly enriched in a group if FDR q <0.2.

## Engrafting PDC into immunodeficient mice

BL avatar. Eight- to 10-wk-old NOD-scid IL2rg null (NSG) mice (The Jackson Laboratory) were injected with BL cell lines (5–10 × 106 cells) subcutaneously into the right flank. We observed palpable tumors within 2–4 wk, depending on the growth rate of the BL cell line. Tumor volume was measured by calipers three times a week and calculated using the modified ellipsoid formula ½(Length*-width2). Criteria for endpoints were tumor burden > 20% of body weight loss (BW), mean tumor volume > 4,000 mm3, and ulceration/necrosis of tumor. Tumor phenotype and EBV content were evaluated by in situ hybridization, immunohistochemistry, and flow cytometry. All animals were housed in a specific pathogen-free facility in microisolator cages and given autoclaved food and

acidified autoclaved water. All animal procedures were done in accordance with the guidelines of the Institutional Animal Care and Use Committee of UMass Chan and conformed to the recommendations in the Guide for the Care and Use of Laboratory Animals (National Institutes of Health).

### Rituximab in vivo experiments

Age-matched mice with both male and female NSG mice were divided into two groups (five–six mice/group) and injected with BL cell lines, as explained above. NSG mice without tumor cell implantation were used as controls. As tumors became palpable, mice were randomized by body weight and injected intraperitoneally (IP) either with rituximab (50 μg/g; Rituxan, Genentech) in 200 μl of PBS or with 200 μl PBS only, every 3 d up to four injections mimicking human treatment regimen of four doses weekly (375 mg/m2). Mice were monitored for adverse clinical signs and euthanized by $CO_2$ asphyxiation followed by cervical dislocation when the tumors exceeded 20% body weight, and were inactive or moribund. Tumor tissues were collected and placed in RNA later (AM7024; Thermo Fisher Scientific) for RNA-seq experiments or harvested and processed, so they could be analyzed by IHC and flow cytometry. Single-cell suspensions of tumor tissues were obtained by mincing tumor tissue using gentleMACS Dissociator (Miltenyi) along with DNase (AMP-D1, 100 μg/ml; Sigma-Aldrich) and hyaluronidase (Cat no.: H6254; Sigma-Aldrich).

### Complement-mediated lysis

BL cell lines were incubated with rituximab (10 μg/ml) for 24 h based on previous studies (Zhao et al, 2013). Rituximab was added in vitro to tumor cell lines (0.5 × 10^6) suspended in complete media supplemented by 10% human serum and inactivated by incubation at 56°C for 30 min. Parallel cultures were suspended in complete media with complement (no heat inactivation).

### Annexin V/7AAD staining

Externalization of phosphatidylserine, an early marker of apoptosis, was monitored by Annexin V-PE and 7AAD staining to measure cell death (PE Annexin V Apoptosis Detection Kit I, 559763; BD Biosciences). BL cell lines treated with rituximab, in vitro, and BL tumor cells harvested from NSG-BL tumors treated with rituximab and PBS were first stained with human antibodies to CD20, CD19, and CD45, followed by Annexin V/7AAD staining. Briefly, the cells were first washed with ice-cold PBS and resuspended in 1X Annexin V binding buffer at $1 \times 10^6$ cells/ml. 100 $\mu l$ of this cell suspension ($1 \times 10^5$ cells) was then incubated with 2.5 μl of Annexin V-PE and 7AAD each, in the dark for 20 min. After incubation, cells were added to 400 μl of 1X Annexin V binding buffer and analyzed by flow cytometry. Isotype IgG antibody was tested as a negative control even though there was no difference between the media-only condition and IgG isotype.

### JC-1 staining

Loss of MMP as a marker of apoptosis was assessed with JC-1 dye (MitoProbe JC-1 Assay Kit, M34152), a cationic carbocyanine dye that accumulates in mitochondria of cells in proportion to MMP. After treatment of the BL cell lines with rituximab in vitro, the cells were harvested, washed, and resuspended in 200 μl of warm medium and stained with JC-1 dye as per the manufacturer's recommendations. Briefly, cells were resuspended at a density of $1 \times 10^6$ cells/ml in warm complete RPMI medium and stained with JC-1 dye at a final concentration of 2 μM for 30 min at 37°C, and analyzed by flow cytometry. Staurosporine (1 μM) was used as a positive control.

### TUNEL assay

Fragmentation of intracellular DNA is a late marker of apoptosis and was measured by TUNEL assay, using the APO-DIRECT Kit (556381; BD Biosciences). Briefly, tumor cells treated with rituximab and PBS were fixed with 1% paraformaldehyde, washed, and stored in ice-cold 70% ethanol for at least 24 h before staining with 50 μl of DNA labeling solution, prepared as per the manufacturer's instructions. After incubation for about 1 h with the DNA labeling solution, the cells were washed and analyzed by flow cytometry.

### Statistical analysis

Statistical analyses were performed using GraphPad Prism 9 and in R version 4.0.3. Statistical analysis was performed by non-parametric $t$ tests using the Mann–Whitney test, or for paired analyses, $t$ test (if normally distributed) or non-parametric Wilcoxon signed-rank test. Differences were considered significant if $P$-value was less than 0.05 (*$P < 0.05$; **$P < 0.005$; and ***$P < 0.0005$).

### Figures

Schemas were created with BioRender.com.

## Data Availability

The raw data generated supporting the results of this article are available and can be accessed in NCBI (National Center for Biotechnology Information) dbGaP (database of Genotypes and Phenotypes) with accession number phs001282.v2.p1.

## Supplementary Information

## Acknowledgements

This study was supported by the U.S. National Institutes of Health, National Cancer Institute, grants R01 CA134051 (AM Moormann), R01 CA189806 (AM Moormann), and R01 CA234348 (MA Luftig). The authors would also like to thank the parents and guardians for enrolling their children in this study. This publication was reviewed and approved by the Scientific and Ethics Review Unit at the Kenya Medical Research Institute.

## Author Contributions

P Saikumar Lakshmi: conceptualization, data curation, formal analysis, investigation, methodology, and writing—original draft, review, and editing.
CI Oduor: conceptualization, data curation, formal analysis, investigation, methodology, and writing—original draft, review, and editing.
CS Forconi: conceptualization, formal analysis, investigation, methodology, and writing—original draft, review, and editing.
V M'Bana: conceptualization, data curation, formal analysis, supervision, investigation, methodology, project administration, and writing—original draft, review, and editing.
C Bly: conceptualization, data curation, formal analysis, supervision, funding acquisition, validation, investigation, methodology, project administration, and writing—original draft, review, and editing.
RM Gerstein: investigation.
JA Otieno: conceptualization, supervision, investigation, methodology, project administration, and writing—original draft, review, and editing.
JM Ong'echa: conceptualization, supervision, funding acquisition, validation, investigation, methodology, project administration, and writing—original draft, review, and editing.
C Münz: conceptualization, resources, data curation, formal analysis, supervision, validation, investigation, methodology, and writing—original draft, review, and editing.
MA Luftig: conceptualization, resources, data curation, formal analysis, supervision, funding acquisition, validation, investigation, visualization, methodology, project administration, and writing—original draft, review, and editing.
MA Brehm: conceptualization, data curation, formal analysis, supervision, investigation, methodology, and writing—original draft, review, and editing.
JA Bailey: conceptualization, resources, data curation, formal analysis, supervision, validation, investigation, methodology, and writing—original draft, review, and editing.
AM Moormann: conceptualization, resources, data curation, formal analysis, supervision, funding acquisition, validation, investigation, visualization, methodology, project administration, and writing—original draft, review, and editing.

## Conflict of Interest Statement

The authors declare that they have no conflict of interest.

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
