## [Reviewer comments · Life Science Alliance]

Life Science Alliance

Endemic Burkitt lymphoma avatar mouse models to explore tumor variation and treatment responses

Priya Saikumar Lakshmi, Cliff Oduor, Catherine Forconi, Viriato M'Bana, Courtney Bly, Rachel Gerstein, Juliana Otieno, John Ong'echa, Christian Münz, Micah Luftig, Michael Brehm, Jeffrey Bailey, and Ann Moormann

DOI: <https://doi.org/10.26508/lsa.202101355>

Corresponding author(s): Ann Moormann, University of Massachusetts Medical School

Review Timeline:

Submission Date:	2021-12-25
Editorial Decision:	2022-02-15
Revision Received:	2022-12-01
Editorial Decision:	2023-01-06
Revision Received:	2023-01-31
Accepted:	2023-02-01

Transaction Report:

February 15, 2022

Re: Life Science Alliance manuscript #LSA-2021-01355-T

Dr. Ann Moormann
University of Massachusetts Medical School
Department of Medicine
Worcester, MA

Dear Dr. Moormann,

Thank you for submitting your manuscript entitled "Patient-derived endemic Burkitt Lymphoma mouse models for exploring inter-patient tumor variation and testing targeted therapies" to Life Science Alliance. The manuscript was assessed by expert reviewers, whose comments are appended to this letter. We invite you to submit a revised manuscript addressing the Reviewer comments.

Thank you for this interesting contribution to Life Science Alliance. We are looking forward to receiving your revised manuscript.

Sincerely,

Eric Sawey, PhD
Executive Editor
Life Science Alliance
<http://www.lsa-journal.org>

B. MANUSCRIPT ORGANIZATION AND FORMATTING:

Reviewer #1 (Comments to the Authors (Required)):

In this manuscript, the authors have established and characterized five new pediatric patient-derived Burkitt cell line and xenograft/avatar models. They are established from male and female patients, occurred in jaw or abdomen, have doubling times in culture of 25-50 hours, have characteristic myc chromosomal translocations, and encompass Type 1 vs 2 EBV. The newly established cell lines express B cell lineage markers, including CD20, CD10, CD19 and the presence of Myc translocation is demonstrated. EBV status is confirmed by EBERs. These are then characterized somewhat by microscopy, though that is shown only for two of the five cell lines, and not for any of the established ones. They are then analyzed by RNAseq of in vitro cultured and after NSG BL avatar passage. PCA and pathway analysis are done and cross compared to established "old" cell lines. A difference spotted is the enrichment of the AFT2-associated gene set in the old set vs the STK33-associated gene set in the new BL cell lines. Apoptosis responses to rituximab are measured. In vivo tumor volumes and survival are measured. IHC is done to look at several latency genes and BZLF1. Tumor volume responses to rituximab are measured. Effects of rituximab on apoptosis induction is measured. One tumor (740) was particularly sensitive to rituximab, and was found to have 878 DE, including IRF7 and several interferon induced genes. While interesting, since this is just from one tumor, it's tough to know the significance.

Overall, the authors create an important new resource and do important studies to characterize these new cell lines. The RNAseq data in vitro and from the avatar models is an important new resource. However, there are important issues to address. In particular, the comparison between old and new is more complicated than just that, as described below, and significantly complicates interpretation of this cross-comparison.

Major:

The latency state of the "old" BL cell lines chosen is an important issue that needs to be addressed. The authors chose "old" cell lines that do not represent the typical Burkitt lymphoma cell lines with type I EBV latency. This could be a strength, but they would have to include typical latency I cell lines. Raji have a near latency III program, but have an EBNA3C deletion as well as a BALF4 deletion. Some Raji episomes are integrated. Daudi are Wp restricted and have EBNA2 deletion. Namalwa is an odd Burkitt cell line, with integrated EBV and some reports claim latency III. Jijoye have latency III. And yet, it is thought that these are all outliers, as most Burkitt tumors are thought to have latency I. So, this is not the set of old Burkitt lymphomas that I would have selected, and adds a variable that has not been fully considered to their PCA analysis and other cross-comparisons. I think the authors should include type I latency Burkitt cell lines in their old group, such as Akata or Mutu. They should also better describe these important facets of the old tumor models in the results and discussion. This is an important additional variable beyond just old versus new that has not been considered.

Were differences in EBV gene expression noted between type I and II tumors? This could be an interesting facet of the paper that was not fully considered.

Would go deeper in the bioinformatics analysis. I feel that the informatic analysis is a good start, but doesn't go deep enough. The authors should also include from their RNAseq a heatmap and spreadsheet showing values of all EBV gene levels. Are the detected EBV miRNAs present at similar levels?

Late apoptotic cells should be Annexin V- and 7AAD+. Annexin V is an early apoptosis marker. This should be corrected.

Is it practical to use rituximab on children with endemic Burkitt? Would comment more in the discussion on the availability of rituximab and what would be needed to implement this in areas of endemic Burkitt. Would rituximab ablation of B-cells be problematic in regions where children are likely to be exposed to malaria?

New but not old cell lines were tested for rituximab sensitivity. Would add data for the old cell lines, to keep the cross-comparison going.

LMP1 levels can be quite low and below the limit of detection. I would want to see clear RNAseq evidence of latency IIb before assigning that latency state. To my knowledge, IIa has not been seen with Burkitt samples, unless perhaps there has been integration of the EBV genome into the host chromosome with deregulated EBV expression.

I'm surprised in Figure 5 that LMP1 is highest in BL740, and absent in BL717. In Figure 1, it was BL717 that grows in clusters that look like Burkitt cells with latency III. More fields for 5F should be shown. How do the authors explain the surprising finding that IRF7 is expressed in BL717 and 720, but not BL740, whereas it is BL740 that highly expresses LMP1? The Pagano group has shown that LMP1 induces IRF7, so this is surprising. I also am surprised that Daudi expresses IRF7 and ISG15. A negative control should be provided to confirm that the stain is properly working.

Minor:

- Would state in the main text what cutoff was used for assignment of differential genes.

-Table 1. Would add patient ages. Would also add info for all of the old EBV cell lines in this table.

-would more clearly state in the introduction that the ~15% of Burkitt tumors with Wp restricted expression have EBV genome deletions that destroy EBNA2 and that truncate EBNA-LP. It is mentioned later in the paper that Wp have EBNA2 deletion. But for the general audience, would put this in the intro.

-would explicitly state in the methods that permission was granted to establish new cell lines, as this is a sensitive issue, given that they were obtained from African children.

Figure 1B-C. Please show representative images from all 5 cell lines, and from the old models for comparison.

-Fig 1E. Please add a square that directs the region on 10X being amplified on 40X

-Fig 2, would define how the cd19, 20 and 10 gates were drawn. In 2D, would add a color legend to the figure itself to enhance readability. In 2E, it would be helpful if the authors can also state the % of EBER+ cells. Since we only see EBER staining with no other cell marker, it's a bit tough to say. Are all cells showing at least some level of EBER staining?

-In 3B, how were these genes chosen to display? Would be best to show the whole volcano so that we can have a fuller picture, and then label genes of interest.

-The E2F23 and HoXA9 pathways were found to be different, but are not described in the results/discussion. Since these are outliers highlighted in the informatic analysis, would discuss these pathways in more detail. Would include in the supplement the list of genes that comprise each of these pathways.

Heatmap data in 5E should be accompanied by median, SEM and statistical significant differences between groups should be indicated. Please state in the legend the number of RNAseq replicates.

Please state how many RNAseq replicates were done. I could not readily find this information in the main text or methods.

Would state how many reads were done per sample.

Would modify the statement on page 7 that "our new cell lines all expressed key BL marker genes including MYC, MS4A1(CD20), MME(CD10), MKI67(KI-67), and BCL6 which distinguishes them from other B cell lymphomas". Most B-cell lymphomas express MYC, CD20, KI-67 and BCL6. CD10 is expressed in germinal center B-cell subtype lymphomas.

For Fig 7 C, higher magnification images should also be shown. Ideally, here and in the other IHC images, analysis software such as Image J should be used to present quantitative comparisons to accompany the representative image shown.

On page 7, would spell out DGE and DE. Would also define 7-AAD and that it is a vital dye.

Would add discussion to why one of the cell lines (BL740) was more susceptible to rituximab alone. Is this related to the admixture of cells growing in the sample (i.e. other PBMCs)?

Would amend the statement on page 8 that bzlf1 is 'sufficient to convert EBV from latent to lytic cycle'. Many cells have BZLF1 but have abortive lytic cycles, with little to no late gene expression.

Scale bars should be added to all IF/IHC images.

Would define PDX on page 12 (the acronym isn't defined). Would state that ibrutinib is a BTK inhibitor (rather than BCR inhibitor)

Reviewer #2 (Comments to the Authors (Required)):

The manuscript by Lakshmi and colleagues attempts to address the development of a PDX model for Burkitt's lymphoma using a mouse model to explore inter-patient tumor variation and to test targeted therapies. They established 5 PDX BL tumor cell lines and the corresponding NGS-BL mouse models and found significant variations in tumor growth and survival among NSG-BL

tumors as well as EBV protein expression patterns. Animals treated with Rituximab also had varying sensitivities, and was identified with IFN α and IRF7 and ISG15 were induced in the animals that were resistant to Rituximab. The author indicate that there are inter-tumor variation and heterogeneity and that the use of BL pdx model are good strategies for therapeutic insights. The study is interesting and if explored can provide new insights into the pathology of eBL for therapeutic interventions. However, there are many areas that lack accuracy and the manuscript overall is more confusing than providing further clarity on the problem. Here are some concerns that should be considered.

1. It is well know that BL are heterogenous in nature and the pattern of EBV gene expression can vary extensively.
2. The use of the term Avatar is misleading and when used in the text is at times distracting. Avatars are descent of a form of deity to earth and is extremely important in some cultures. Therefore, the use in a superfluous or negative manner (in relationship to cancer) can be taken as an insult.
3. The use of Rituximab alone if misleading and BL is mostly treated with CHOP and R-CHOP when it is included in the regimen. Variation in CD20 in B cell cancers are well known and documented as to escape from Rituximab for many refractory B cell cancers. Hence, most will have combination therapy in addition to Rituximab.
4. Many papers are not cited in the literature and the authors have cited their review or own work with disregard for other seminal studies related to BL.
5. The type of latency in EBV is one of the most complicated in the literature and now with variations of types with a and b lettering only adds to the confusion. Please disregard the nomenclature as it is not established by the field and not be pushed to set additional complications to what is already a difficult concept. Anyone in the field would know that it is best to reduce complexity than increase for use of dogma.
6. The variations in EBV gene expression based on the provided data is based on sequencing that will have differences based on depth and the percentage of cells infected with the virus in tumors. Saying there are differences without establishing a baseline to normalize expression is incorrect and at least should be followed up by validation of all the transcripts using RT-qPCR as well as IHC quantitation.
7. The establishment of the cultures before implantation is a potential problem as within the window of 7-10 days or more cell duplication can introduce new mutational burden that is distinct from the parental tumor. Why not implant the tumor directly without expansion to reduce this possibility as it makes interpretation of the data more complicated and potential for mistakes. It was also not clear how many total days of expansion before implantation?
8. There is a mistake in figure 2 reference in the text for CD10 which is CD20 in the figure 2C.
9. EBER signals fro ISH is fuzzy in some of the pictures and the use of droplet PCR for copy number is difficult to reconcile as the standard curve was not determined or established based on the methods described. Missing negative control 40x as well as no specific probe control for EBER staining
10. How was the 2 types of EBV determined by droplet PCR as both EBNA2 and EBNA3C will have varying sizes and its not clear how that would be identified using this methodology.
11. Some of the analyses and the description of the results seems discordant to what is shown in the figures themselves as if the authors have not studied the data properly.
12. The identification of some of the genes in figure 3. The PCA and volcano plots shows discrepancies between old and new BL cell lines but stated as similar in text.
13. How are these genes STK33, MEK, ATF2 and HoxA9 differ in previous reports of BL sequencing in the data bases. Is this a unique finding? Maybe a heat map would be advantageous.
14. Tumor take of 2/6 if low not moderate and represents implantation and not tumour burden.
15. Stating that LMP1 was only expressed in one BL tumor is a bit of a reach as it should be validated for sensitivity and sequencing depth concerns that can arise with the RNA-Seq.
16. The uniformity of the IHC staining in counterstain should be balanced and controls provided. Why did they use Daudi here as a control of Figure 5 instead of a well know BL cell line.
17. Signals for EBNA1 is lower than EBNA2 for 5E which is worrisome as EBNA1 should be most prominent signal for BL cells/tumors.
18. Some of the supplementary figures are unreadable and so not useful. Difficult to understand what the authors were trying to show when one cannot read the figures.

Overall: This is a potentially important study but the manuscript lacks careful analysis and experimental design, which makes interpretation of the data problematic. At this point there is not much more information that this manuscript provides that is not already know in the literature and at times complicates the field.

We would like to thank the reviewers for their time and instructive comments, and for the extension granted to submit our revised manuscript in order to include additional experiments, as suggested.

In summary our revised manuscript includes the following major changes:

1) We have sequenced and now include “typical” type I latency BL cell lines, Mutu and Akata in our analyses of old versus new BL cell lines in Figure 3 and Figure 5D. Their inclusion does not alter the conclusion from our initial findings and further demonstrates that old BL cell lines significantly differ from our new BL cell lines and corresponding eBL patient tumors.

2) We performed additional experiments (in triplicate) measuring *in vitro* rituximab-induced cell death to include Mutu and Akata BL cell lines that now appear in revised Figure 4B. Our results show that only one of our new BL cell lines (BL740) is similar to Akata and Mutu, being sensitive to direct killing by rituximab (with Mutu and Akata being even more sensitive in the presence of complement). Four of our new BL cell lines appeared to be resistant to direct killing by rituximab. In addition, loss of mitochondrial potential only appeared to be significant for Mutu and BL740 whereas the other lines remained unchanged by rituximab treatment. Together, these experiments reinforce one of the main points of our manuscript, that these older BL cell lines are not representative of patient tumors and could result in misleading pre-clinical results when testing new therapies. Thus, the translational potential of old BL cell lines may be limited compared to newly-established BL tumor cell lines.

3) In order to more comprehensively characterize EBV latency patterns within NSG-BL avatar tumors, we expanded the IHC stains to include EBNA3A and EBNA3C (which were back ordered from last year due to the COVID-19 pandemic shutdown). We were also able to obtain a negative control NSG-mouse melanoma tumor (AV21.91) and used the workflow illustrated in Figure 1 to generate NSG-Mutu tumors (Akata cell lines failed to establish tumors within this mouse model). The IHC images (20x and 40x) have been moved to supplemental Fig S6. Using Qu-Path software we quantified viral protein expression, summarized in a new Figure 5E. The patterns and interpretations have been added to the Results and Discussion sections. (We also used Qu-Path to quantify IHC results in Figure 7E.) These additional results support our initial conclusion that our new BL cell lines generate tumors with diverse latency patterns and varying degrees of lytic reactivation. The implications of this finding is explored in our Discussion.

4) Since this manuscript was submitted Dr. Priya Lakshmi completed her PhD in Jan 2022. A new post-doc, Dr. Viriato M'Bana was hired in the Moormann lab in Feb 2022. With the assistance of Dr. Lakshmi, Dr. M'Bana conducted the additional experiments and has been added to the revised manuscript as a co-author. We are also including Courtney Bly as a co-author. She optimized and conducted the additional IHC experiments and the Qu-Path analyses that appear in Figures 5 & 7. They have both contributed to writing this revised manuscript.

Point-by-point responses to each comment appear in blue below.

Reviewer Comments:

Reviewer 1:

Reviewer #1 (Comments to the Authors (Required)):

In this manuscript, the authors have established and characterized five new pediatric patient-derived Burkitt cell line and xenograft/avatar models. They are established from male and female patients, occurred in jaw or abdomen, have doubling times in culture of 25-50 hours, have characteristic myc chromosomal translocations, and encompass Type 1 vs 2 EBV. The newly established cell lines express B cell lineage markers, including CD20, CD10, CD19 and the presence of Myc translocation is demonstrated. EBV status is confirmed by EBERs. These are then characterized somewhat by microscopy, though that is shown only for two of the five cell lines, and not for any of the established ones. They are then analyzed by RNAseq of in vitro cultured and after NSG BL avatar passage. PCA and pathway analysis are done and cross compared to established "old" cell lines. A difference spotted is the enrichment of the AFT2-associated gene set in the old set vs the STK33-associated gene set in the new BL cell lines. Apoptosis responses to rituximab are measured. In vivo tumor volumes and survival are measured. IHC is done to look at several latency genes and BZLF1. Tumor volume responses to rituximab are measured. Effects of rituximab on apoptosis induction is measured. One tumor (740) was particularly sensitive to rituximab, and was found to have 878 DE, including IRF7 and several interferon induced genes. While interesting, since this is just from one tumor, it's tough to know the significance. Overall, the authors create an important new resource and do important studies to characterize these new cell lines. The RNAseq data in vitro and from the avatar models is an important new resource. However, there are important issues to address. In particular, the comparison between old and new is more complicated than just that, as described below, and significantly complicates interpretation of this cross-comparison.

Major:

Comment R1.1: The latency state of the "old" BL cell lines chosen is an important issue that needs to be addressed. The authors chose "old" cell lines that do not represent the typical Burkitt lymphoma cell lines with type I EBV latency. This could be a strength, but they would have to include typical latency I cell lines. Raji have a near latency III program but have an EBNA3C deletion as well as a BALF4 deletion. Some Raji episomes are integrated. Daudi are Wp restricted and have EBNA2 deletion. Namalwa is an odd Burkitt cell line, with integrated EBV and some reports claim latency III. Jijoye have latency III. And yet, it is thought that these are all outliers, as most Burkitt tumors are thought to have latency I. So, this is not the set of old Burkitt lymphomas that I would have selected and adds a variable that has not been fully considered to their PCA analysis and other cross comparisons. I think the authors should include type I latency Burkitt cell lines in their old group, such as Akata or Mutu. They should also better describe these important facets of the old tumor models in the results and discussion. This is an important additional variable beyond just old versus new that has not been considered. Were differences in EBV gene expression noted between type I and II tumors? This could be an interesting facet of the paper that was not fully considered.

Response: We agree with the reviewer that comparing our new BL cell lines with classical latency I BL cell lines such as Akata and Mutu strengthens our manuscript. Thus, our analyses in Figure 3 and gene expression analysis (Fig 5D) now include additional RNAseq sequencing of traditionally used latency I, 'old' BL cell lines-- Akata and Mutu. Adding these lines did not change our results but only strengthened them. We have also performed additional in vitro rituximab experiments with these lines measuring apoptosis and mitochondrial potential that now appear in revised Fig 4D which showed that our BL740 cell line behaves more like Mutu than Akata, while the 4 other new BL cell lines do not. Akata and Mutu are also included in Figure 4D, demonstrating variability across BL cell lines that should be appreciated when testing rituximab or other treatments. We also injected Mutu and Akata cell lines into our NSG mouse models but found that Akata would not establish tumors whereas Mutu did. Figure 5E now includes EBV protein expression from NSG-Mutu tumors. However, there was insufficient funds (and time) to include additional NSG mouse model experiments using rituximab and these older BL cell lines. Our next manuscript will include Mutu alongside our new BL cell lines but in an NSG mouse model with reconstituted human immune cells which is a better model to test

immunotherapies that involve antibody-dependent cellular cytotoxicity. We thank the reviewer for this very helpful suggestion.

We have included a fuller discussion of the old cell lines (which were derived from the same geographic area in Africa as our new BL cell lines) in our revised manuscript.

“BL cell lines established over 50 years ago, such as Daudi (Klein et al. 1968), Jijoye (Kohn et al. 1967), Namalwa and Raji (Pulvertaft 1965), and more recently Akata (Takada et al. 1991) and Mutu (Gregory et al. 1990) established the dogma that EBV in BL tumors are latency I (Rowe et al. 1987). However,...”

With regards to the EBV latency, the reviewer claims all BL cell lines are considered to be latency I. The classical phenotype of BL was considered to be latency I, but there is evidence of promiscuity in EBV gene expression from Beverley Griffin’s group and heterogeneity within and across BL lines from primary tumors (Xue et al. 2002; Kelly et al. 2002; Kelly et al. 2006). We have established new BL cell lines from tumor biopsies that show (in this manuscript) that there is expression of other EBV proteins such as EBNA2, BZLF1 in a small subpopulation within the BL cell lines, in support of EBV latency being more dynamic.

We decided to remove the EBV gene expression data (previous manuscript, Figure 5E) because the methods used did not enrich for viral transcripts which are very GC rich. We plan to explore viral gene and protein expression with the context of our BL cell lines and NSG BL tumor models in more depth in another manuscript. Therefore, we have included the following in the Discussion of the revised manuscript:

“Characterizing and interrogating differences in EBV gene and protein expression between type 1 and type 2, and abortive versus productive lytic reactivation, are complex questions that will be addressed in subsequent studies. We also suspect the lack of host immune pressure in our model system is permissive of more viral protein expression, thus future studies will focus on understanding what mechanisms are involved in allowing this variation.”

Comment R1.2 Would go deeper in the bioinformatics analysis. I feel that the informatic analysis is a good start, but doesn't go deep enough. The authors should also include from their RNAseq a heatmap and spreadsheet showing values of all EBV gene levels. Are the detected EBV miRNAs present at similar levels?

Response: We would like to point out that our RNAseq libraries were prepared using the poly-A enrichment of mRNA molecules from Total RNA extracted from each sample. Therefore, this experiment was not designed to capture human or viral miRNA expression. Details of this procedure have been mentioned in the method section of the manuscript, page 18, section “Library preparation and sequencing”. We have however reported miRNA comparisons directly from BL patient tumors in a separate publication (Oduor, et al., 2017). In our previous publication EBV miRNAs made up 2.7% of the miRNA composition in the eBL samples. However, we did not find any significant associations regarding initial patient outcome or anatomical presentation. As mentioned above, deeper interrogations into the role and kinetics of EBV gene expression in eBL pathogenesis, tumor maintenance and treatment response are the focus of a subsequent manuscript.

Comment R1.3 Late apoptotic cells should be Annexin V- and 7AAD+. Annexin V is an early apoptosis marker. This should be corrected.

Response: We disagree with the reviewer's interpretation. Following the methodology paper, (Vermes et al. 1995) and the manufacturer of the BD kit we used, early apoptotic cells are characterized as Annexin V+, 7AAD- and late apoptotic cells are Annexin V+ 7AAD+. This nomenclature has now been clarified in the results section of the revised manuscript:

“Rituximab-mediated direct killing was measured by detecting phosphatidylserine (PS) exposure via AnnexinV and 7AAD staining (Figure 4A). Early apoptotic cells are AnnexinV+ 7AAD- (Vermes et al. 1995) whereas late apoptotic cells, as well as necrotic cells are characterized by both AnnexinV+ and 7AAD+ staining. We found that the new BL cell line, BL740, showed significant cell death based on

AnnexinV+7AAD+ staining (33% vs 18% media alone, p-value = 0.04), as a result of direct rituximab activity whereas the other four new BL cell lines were not affected (Figure 4B)."

Details on the BD kit used to identify the late apoptotic cells have been included in the method section, Page 20. "**Annexin V-7AAD staining.** Externalization of phosphatidylserine, an early marker of apoptosis, was monitored by using AnnexinV-PE and 7AAD staining to measure cell death (PE Annexin V apoptosis detection kit 1, BD Biosciences, 559763)."

Comment R1.4 Is it practical to use rituximab on children with endemic Burkitt? Would comment more in the discussion on the availability of rituximab and what would be needed to implement this in areas of endemic Burkitt. Would rituximab ablation of B-cells be problematic in regions where children are likely to be exposed to malaria?

Response: Rituximab is already being used in Africa to treat children, but in limited settings. Rituximab was added to the chemotherapeutic regimen for children diagnosed with CD20+ lymphoma in Eldoret, Kenya in 2020. This was done only after strengthening the oncology-specific training and supportive care of the medical staff and pharmacists. We agree that there are concerns about the use of rituximab for children at risk of exposure to infectious diseases such as malaria, and who still require healthy CD20+ B cells to make protective antibodies. In general rituximab is thought to be more benign than higher-dose chemotherapy that can lead to severe neutropenia. Infectious risks do not seem to be greater in those treated with rituximab for malignancy. However, infectious risks are significantly increased in those requiring chronic treatment with rituximab such as rheumatologic patients. For children who develop eBL and require a short-course of treatment, the patients tend to be 5-9 years of age and have been exposed to malaria since infancy, and have significant preexisting antibodies and antibody-producing plasma B cells (that no longer express CD20). Subsequent to the treatment the naive B cells from the marrow should be able to respond appropriately although there is a possibility that if antibodies wane their infection could be more severe. As part of our study protocol, we continue to collect monthly blood samples from eBL children for 2-years post-diagnosis and to determine all cause mortality outcomes. We plan to assess the maintenance of preexisting antibody levels to a panel of malaria antigens by Luminex, as well as measure the induction of new antibodies to variant malaria antigens and other infections or vaccines over the 2-year period after eBL diagnosis. We have expanded our discussion in this manuscript to mention this concern and what measures could be put into place to decrease the risk of infectious diseases for these children until their immune system has recovered.

"Our study is timely because rituximab, a standard of care combined with chemotherapy for BL treatment in high-income countries, is being introduced within pediatric eBL treatment regimens along with strengthening supportive care for oncology patients in LMICs, including Kenya. Understanding rituximab failures within this context, including possible increased susceptibility to endemic infectious diseases, will be another important step toward improving overall outcomes for these children".

Comment R1.5 New but not old cell lines were tested for rituximab sensitivity. Would add data for the old cell lines, to keep the cross-comparison going.

Response: We thank the reviewer for this suggestion. We have included Akata and Mutu in our *in vitro* rituximab experiments however, we could not afford to perform additional NSG-mouse model experiments. Their addition would not influence the key findings we present on our new BL cell lines, that they are variable across patients and are closer to the status of the patient tumors than old cell lines.

Comment R1.6 LMP1 levels can be quite low and below the limit of detection. I would want to see clear RNAseq evidence of latency IIb before assigning that latency state. To my knowledge, IIa has not been seen with Burkitt samples, unless perhaps there has been integration of the EBV genome into the host chromosome with deregulated EBV expression.

Response: We agree that viral gene expression levels may be too low for detection, especially considering we did not design our methods to enrich for viral gene expression. Therefore, we removed the old Figure 5E (gene expression from our bulk RNAseq analysis) and plan to specifically explore viral gene expression patterns and

possible EBV genome integration into the host chromosomes in another manuscript. However, the latency patterns of our NSG-BL tumors were able to be assigned based on viral protein expression from the IHC experiments (new Figure 5E). As mentioned in another comment, we also expanded our IHC panel to include antibodies to other latent proteins (EBNA 3A and EBNA3C) that were back-ordered during the COVID-19 pandemic. Based on these results, NSG-BL tumors express the following latency patterns (keeping in mind these are in the absence of a human immune cell reconstitution, which is a future direction and outside the scope of this study). Using Qu-Path, we were able to more accurately quantify viral protein expression in our NSG-BL avatar tumors (new Figure 5E).

“We found that only BL719 and Mutu displayed the classic latency I pattern (EBNA1 only) associated with BL. In addition, we measured LMP1 (latency protein expressed in latency II and III) and EBNA-2 (deleted in Wp-restricted latency but expressed in latency III). We found LMP1 only expressed in BL740, yet with low expression of EBNA2 suggesting a latency IIa/III pattern (Price and Luftig 2015). Inclusion of EBNA-2 expression for BL717, BL720 and BL725 would define them as latency IIb (Price and Luftig 2015). Daudi cell lines have been previously reported to express EBNA3C *in vitro* (Jones et al. 1984). This suggests that the viral protein expression is not fixed throughout the mouse tumors and varies in the absence of immune pressure.”

Figure 5E now shows viral protein expression quantified using Qu-Path software which recommends 20x resolution. We have included IHC representative images at 20x for each tumor and antibody combination in a new supplemental Figure S6A. We have moved the 40x resolution images to supplemental Figure S6B.

Comment R1.7 I'm surprised in Figure 5 that LMP1 is highest in BL740, and absent in BL717. In Figure 1, it was BL717 that grows in clusters that look like Burkitt cells with latency III.

Response: First, we would like to clarify the point we are trying to make in Figure 1B, that if the cells cluster or not during the establishment of a new BL cell line should not be used as a criteria to determine success of culturing. We also successfully modified the tumor cell culturing method from Georg Bornkamm so feeder cells were not necessary. Second, this phenotype is not necessarily stable and may change after cell lines have been established or after several passages. We did not measure LMP1 expression during the initial establishment of each new cell line. However, in our hands BL717 continues to grow in clusters and forms NSG-BL tumors even with no LMP1 expression by IHC (now quantified more accurately using Qu-Path). Whereas, the BL719 cell line once established can form clumps when grown to high confluency, although the doubling time of this cell line is by far the slowest (Table 1). Due to these observations, we are conducting kinetic studies to measure changes in viral gene and protein expression patterns within each cell line *in vitro*. These results will be featured in another manuscript focusing on the virus within the context of our new BL cell lines. Third, protein expression shown in Figure 5 is from NSG-BL tumors (not cell lines) in a mouse model that does not have reconstituted human immune cells and therefore is not under immune pressure. We have reviewed the manuscript and have tried to clarify these distinctions (and context) in the results and discussion.

Comment R1.8 How do the authors explain the surprising finding that IRF7 is expressed in BL717 and 720, but not BL740, whereas it is BL740 that highly expresses LMP1? The Pagano group has shown that LMP1 induces IRF7, so this is surprising. I also am surprised that Daudi expresses IRF7 and ISG15. A negative control should be provided to confirm that the stain is properly working.

Response: As the reviewer mentions Joe Pagano's (Ning et al. 2003, Zhang et al. 2001) and Elliott Kieff's (Song et al. 2008) groups compared latency III to latency I BL lines and showed that IRF7 was always expressed in the latency III lines with LMP1. LMP1, an oncoprotein of EBV, activates IRF7 through a RIP- and TRAF6-dependent ubiquitination pathway. Inactive IRF7 is present in the cytoplasm and viral infection triggers translocation into the nucleus to activate transcription. To further understand this, we measured ISG15 (Interferon-stimulated gene 15) which is downstream of the IRF7 signaling pathway, and which was also not expressed. More research will be necessary to understand why LMP1 expression does not induce IRF7 in BL740. We also injected Mutu and Akata cell lines into NSG mice (revised Figure 7D and 7E) and found that

Akata was not able to produce tumors in our NSG mouse model however Mutu did. We found NSG-Mutu tumors expressed ISG15 but not IRF7 (similar to BL740). Interrogating the regulation of these pathways is mentioned in the revised Discussion as a future direction of investigation.

We have now included a negative control tumor AV21.91 (melanoma grown in the same NSG mouse avatar background) which is clearly negative for both IRF7 and ISG15. We confirmed with multiple tumor sections that Daudi tumors grown in NSG mice express IRF7 and ISG15 (~50%).

Minor:

Comment R1.9 Would state in the main text what cutoff was used for assignment of differential genes.

Response: The cutoff used to assign differentially expressed genes was a log₂ Fold change of 1.5, p-value < 0.01 and false discovery rate (FDR) < 0.01. This has been outlined in the revised manuscript method section “Gene expression analysis” section, page 19, “*To control for multiple testing, we applied the Benjamini-Hochberg procedure (BH) with a threshold for statistical significance set at an adjusted p-value < 0.01 and false discovery rate (FDR) < 0.01.*”

Comment R1.10 Table 1. Would add patient ages. Would also add info for all the old EBV cell lines in this table.

Response: We thank the reviewer for this comment. The ages of our patients have been added to the revised Table 1 (page 28) of the revised manuscript. Information on the old BL cell lines are publicly known since they are all commercially available. We have referenced the sources and what is known of the clinical and molecular characteristics of the old BL cell lines in the text but have not included them in Table 1 because they are not our primary data and information on them are at times incomplete.

Comment R1.11 -would more clearly state in the introduction that the ~15% of Burkitt tumors with Wp restricted expression has EBV genome deletions that destroy EBNA2 and that truncate EBNA-LP. It is mentioned later in the paper that Wp have EBNA2 deletion. But for the general audience, would put this in the intro.

Response: We thank the reviewer for this suggestion. We have edited the manuscript to mention this in the Introduction.

“EBV-associated tumors generally display more restricted latency programs as an immune evasion strategy (Shannon-Lowe and Rickinson 2019); with eBL tumors being classically categorized as EBV latency I, only expressing EBV nuclear antigen 1 (EBNA1), EBV encoded small RNAs (EBERs) and BART miRNAs (Rowe et al. 1986). However, more in-depth studies found 15% BL cell lines displayed Wp-restricted latency where EBNA-2 is deleted, but the W promoter is active enabling expression of EBNA1, 3A, 3B, 3C, and BHRF1 (Kelly et al. 2013) and another study of eBL tumors in Malawian patients showed promiscuity in the latency pattern with the expression of lytic cycle genes (Xue et al. 2002; Labrecque et al. 1999).”

Comment R1.12 -would explicitly state in the methods that permission was granted to establish new cell lines, as this is a sensitive issue, given that they were obtained from African children.

Response: As stated in the manuscript, ethical approval was obtained for this study which includes all aspects therein presented, including the generation of tumor cell lines for research and not for commercial purposes. As additional protection of human subjects, KEMRI has a manuscript approval process to make sure what is being published is consistent with the approved protocol (as stated in the acknowledgements). What the reviewer is requesting is unconventional. Therefore, we consulted IRB authorities in Kenya and at UMass. They agreed that this explicit statement is not necessary in a publication, as it could also apply to other sensitive issues such as saving blood which could be used for genetic studies. If the Editors would like to contest this decision not to change the ethical statement in our manuscript, then we are open to another

conversation with the ethical review committees overseeing this project and resulting publications.

Comment R1.13 Figure 1B-C. Please show representative images from all 5 cell lines, and from the old models for comparison.

Response: We have decided not to change Figure 1 since it would not add any new information to the paper. The purpose of including these images is to reassure those replicating our culturing method that a clustering phenotype should not be used to determine if the cell line established is BL. This has been discussed in more detail in the response above.

Comment R1.14 -Fig 1E. Please add a square that directs the region on 10X being amplified on 40X

Response: We believe the reviewer was referring to Figure 2E. We have added a square in the 10x image indicating from where the 40x image was amplified.

Comment R1.15 -Fig 2, would define how the cd19, 20 and 10 gates were drawn. In 2D, would add a color legend to the figure itself to enhance readability. In 2E, it would be helpful if the authors can also state the % of EBER+ cells. Since we only see EBER staining with no other cell marker, it's a bit tough to say. Are all cells showing at least some level of EBER staining?

Response: We have added FMO (Fluorescence minus one) controls as well as unstained controls to show how the gates were drawn in Figure 2A. We used the program Qu-Path to quantify % EBER positive for each cell line which is now included in Figure 2E and in the results section.

Comment R1.16 -In 3B, how were these genes chosen to display? Would be best to show the whole volcano so that we can have a fuller picture, and then label genes of interest.

Response: We appreciate this suggestion. Figure 3B is an MA plot and has been revised to show all the genes differentially expressed between the new BL cell lines and the patient eBL tumors. Selected known BL marker genes that have been identified in previous BL publications and shown to be highly expressed in BL tumor cells, are highlighted in blue text in the revised Figure 3B. The MA plot shows that there is no significant difference in the expression of these BL marker genes in the new eBL cell lines, compared to the eBL patient tumors. These details have been outlined in the legend of Figure 3B, which now states:

“MA plot showing genes differentially expressed between the new BL cell lines and their corresponding patient eBL tumors. The MA plot illustrates a log₂FC (Fold change) in the genes expressed against the average normalized expression of these genes. Only 456 genes were identified to be differentially expressed of the 11,235 genes tested (Supplementary Table S2). The red and blue dotted lines represent logFC cut-offs of 1.5 and -1.5 respectively. The gene names highlighted blue text in the figure represent known BL marker genes, such as MS4A1 (CD20); MME (CD10); AICDA (AID); MKI67 (Ki-67), which were not differentially expressed between the new cell lines and the patient tumors.”

Comment R1.17 -The E2F23 and HOXA9 pathways were found to be different, but are not described in the results/discussion. Since these are outliers highlighted in the informatic analysis, would discuss these pathways in more detail. Would include in the supplement the list of genes that comprise each of these pathways.

Response: Thank you for this suggestion. The list of genes enriched for these pathways have been added to the revised supplemental tables document (Table S4). The HOXA9 pathway together with MYC is known to help maintain the expression of multiple anti-apoptotic pathways to promote tumor development (Miyamoto et al. 2021). However its role in BL has not been well described. With regards to these results we think it may be aiding in maintaining the survival of the new BL cell lines in the presence of MYC's overexpression. This has been mentioned in the discussion as:

“We also observed that genes associated with STK33 and HOXA9 upregulation were relatively enriched in new BL cell lines. STK33 phosphorylates pro-apoptotic BAD (BCL-xL) and suppresses apoptosis via mitochondrial pathways (Scholl et al. 2009), while HOXA9 together with MYC, is known to

help maintain the expression of multiple anti-apoptotic pathways (Miyamoto et al. 2021) that could improve survival of the cell lines.”

Comment R1.18 Heatmap data in 5E should be accompanied by median, SEM and statistical significant differences between groups should be indicated. Please state in the legend the number of RNAseq replicates. Please state how many RNAseq replicates were done. I could not readily find this information in the main text or methods. Would state how many reads were done per Sample.

Response: Regarding Figure 5E in the initial manuscript, the heatmap shows the log2 transformed expression values of the EBV genes identified from the RNAseq experiments to try and describe the EBV expression patterns within the patient tumors, cell lines and the NSG mouse tumors. For each mouse tumor biopsy and eBL cell line we had 3 technical replicates, sequenced at an average depth of 10 million reads per sample replicate. The average sequencing read depth has been added to the revised method section. However, upon further examination of our methods which did not enrich for viral gene expression (as mentioned above), we have decided to remove this figure and these results from the revised manuscript. Fully interrogating viral gene expression within BL cell lines and changes that may occur from *ex vivo* patient tumor analyses (by scRNAseq) to potential changes in latency and lytic gene expression during culture adaptation and passaging is now the focus on a subsequent manuscript characterizing the virus within these new cell lines and will be compared to traditional cell lines and published studies.

Comment R1.19 Would modify the statement on page 7 that "our new cell lines all expressed key BL marker genes including MYC, MS4A1(CD20), MME(CD10), MKI67(KI-67), and BCL6 which distinguishes them from other B cell lymphomas". Most B-cell lymphomas express MYC, CD20, KI-67 and BCL6. CD10 is expressed in the germinal center B-cell subtype Lymphomas.

Response: Thank you for pointing out this overstatement. We have revised our statement to read “our new cell lines all expressed key genes used to diagnose BL, including MYC, MS4A1(CD20), MME(CD10), MKI67(KI-67), and BCL6.”

Comment R1.20 For Fig 7 C, higher magnification images should also be shown. Ideally, here and in the other IHC images, analysis software such as Image J should be used to present quantitative comparisons to accompany the representative image shown.

Response: *Both Figures 7C and 7D are now shown at 40x. We used the Qu-Path program to quantify expression of IRF7 and ISG15 and have included their respective frequencies in Figure 7E.*

Comment R1.21: On page 7, would spell out DGE and DE. DGE (Differential gene expression)

Response: We apologize for this oversight and the manuscript has been revised accordingly. Differential gene expression (DGE) and differentially expressed (DE) genes are now defined.

Comment R1.22: Would also define 7-AAD and that it is a vital dye. Would add discussion to why one of the cell lines (BL740) was more susceptible to rituximab alone. Is this related to the admixture of cells growing in the sample (i.e. other PBMCs)?

Response: Since 7-ADD is a commonly used reagent to assess cell death so we defined it in the methods section. At this stage of our investigations into the variability between patient-derived cell lines, we do not know why BL740 is more susceptible to rituximab alone. However, we do discuss possible reasons and future directions to figure it out in our manuscript.

To address the question of purity of our BL cell lines, we now include a new supplemental figure S7 demonstrating by CD3, CD19 and CD20 staining and flow cytometry that they are all B cells. We also clarify in the methods that the use of cyclosporin A (CSA) in the media used to establish the BL cell lines interferes with IL-2 synthesis and does not allow for growth of T cells. Since the cell lines are derived from tumor biopsies and not PBMCs, it is highly unlikely other cells besides tumors would continue to grow under such conditions.

Comment R1.23: Would amend the statement on page 8 that bzlf1 is 'sufficient to convert EBV from latent to the lytic cycle'. Many cells have BZLF1 but have abortive lytic cycles, with little to no late gene expression.

Response: The reviewer makes an excellent point about productive lytic versus abortive lytic cycles. We have revised this statement and will be exploring this end point in the EBV-focused manuscript we have mentioned.

Comment R1.24: Scale bars should be added to all IF/IHC images

Response: We apologize for missing this detail. Scale bars have been added to all IHC images.

Comment R1.25: Would define PDX on page 12 (the acronym isn't defined). Would state that ibrutinib is a BTK inhibitor (rather than BCR inhibitor).

Response: Abbreviations have been defined. We apologize for the oversight.

Reviewer #2 (Comments to the Authors (Required)):

The manuscript by Lakshmi and colleagues attempts to address the development of a PDX model for Burkitt's lymphoma using a mouse model to explore inter-patient tumor variation and to test targeted therapies. They established 5 PDX BL tumor cell lines and the corresponding NGS-BL mouse models and found significant variations in tumor growth and survival among NSG-BL tumors as well as EBV protein expression patterns. Animals treated with Rituximab also had varying sensitivities, and was identified with IFN α and IRF7 and ISG15 were induced in the animals that were resistant to Rituximab. The author indicate that there are inter-tumor variation and heterogeneity and that the use of BL pdx model are good strategies for therapeutic insights. The study is interesting and if explored can provide new insights into the pathology of eBL for therapeutic interventions.

However, there are many areas that lack accuracy and the manuscript overall is more confusing than providing further clarity on the problem. Here are some concerns that should be considered.

Comment R2.1: It is well know that BL are heterogenous in nature and the pattern of EBV gene expression can vary extensively.

Response: We apologize for not making a clearer statement about the context for differences in viral gene expression profiles. The following discussion has been added to the Discussion section of the revised manuscript: "The dogma is that BL patient tumors are always latency I and therefore 'homogeneous' in nature. However, when the EBV-associated tumor cells are grown *in vitro* or in a mouse model, the immune pressure limiting viral protein expression is alleviated, thus allowing latency patterns to change as more viral proteins are expressed. To address this point, we have included studies published by Beverly Griffin who was the first to characterize differences in viral protein expression within patient tumors from Malawi, which challenges the dogma of BL being latency I. In addition, we have noted that heterogeneity ranges from the Wp-restricted lines (Kelly et al. 2002) to a host of sub-clonal expression patterns (Kelly et al. 2006). Moreover, recent work by Ben Gewurz and colleagues indicate that de-repression of viral gene expression (both latent and lytic) can occur in the context of methionine restriction both *in vitro* and in xenografts of BL lines (Guo et al. 2022). These studies collectively paint a more heterogeneous picture for the BL transcriptome than was previously appreciated. In fact, recent single cell RNA seq studies (SoRelle et al. 2021; Bristol et al. 2022) are now illuminating the extensive diversity of cell phenotypes in EBV-infected B cells and BL very likely will display a similar breadth of phenotypes when these tumors and cell lines are profiled."

Comment R2.2: The use of the term Avatar is misleading and when used in the text is at times distracting. Avatars are descent of a form of deity to earth and is extremely important in some cultures. Therefore, the use in a superfluous or negative manner (in relationship to cancer) can be taken as an insult.

Response: We have no intention of offending anyone but we did not coin the term 'avatar' mouse models. Since this is now an established term and used to search for comparable studies of human tumors grown in

NSG mice, we chose to follow the nomenclature being used for humanized mouse models of cancer. (<https://www.nature.com/articles/d41586-018-06982-1>, https://en.wikipedia.org/wiki/Mouse_avatar). Additionally, we did not find any significant controversy surrounding the use of the term in our searches and it is a long-standing term in computer science ([https://en.wikipedia.org/wiki/Avatar_\(computing\)](https://en.wikipedia.org/wiki/Avatar_(computing))). If this word is inappropriate within the context of cancer mouse models, then we would certainly support shifting to a new term after a general discussion within the community at large. However, for the sake of consistency within this growing field and for our manuscript to be found using this now common search word, we would like to retain its usage for now.

Comment R2.3: The use of Rituximab alone is misleading and BL is mostly treated with CHOP and RCHOP when it is included in the regimen. Variation in CD20 in B cell cancers are well known and documented as to escape from Rituximab for many refractory B cell cancers. Hence, most will have combination therapy in addition to Rituximab.

Response: We apologize for not being clear as we were not suggesting it would be a monotherapy and wholeheartedly agree with the reviewer that patients would be treated with a combination of conventional chemotherapy and immunotherapies. We do not believe experiments presented in this paper are therefore misleading because they demonstrate that variation in responsiveness across tumors can be shown to be isolated to rituximab - which has implications for dissecting the magnitude of treatment responses since there are several mechanistic pathways by which rituximab can kill tumor cells. As we state in the conclusion, the establishment of this model system will allow us to now delve further into targeted as well as combination therapies in our future studies.

Comment R2.4: Many papers are not cited in the literature and the authors have cited their review or own work with disregard for other seminal studies related to BL.

Response: We apologize for unintentionally not citing seminal and other relevant studies related to BL. We have reviewed the literature, added more references and have included a more extensive discussion on how our findings fit into the field.

Comment R2.5: The type of latency in EBV is one of the most complicated in the literature and now with variations of types with a and b lettering only adds to the confusion. Please disregard the nomenclature as it is not established by the field and not be pushed to set additional complications to what is already a difficult concept. Anyone in the field would know that it is best to reduce complexity than increase for use of dogma.

Response: While we agree with this reviewer that clarity is best, we also recognize that the biology of EBV latent infection is complex. The expression of EBNA2 (and other EBNA) in the absence of LMPs has been described in early infection (Price and Luftig 2014; Price et al. 2012; Messinger et al. 2019; Price and Luftig 2015), EBV infection in humanized mice (McHugh et al. 2017), and EBV infection of CLL cells (Klein et al. 2013). The cyclical expression of LMP1 in latency III cells does not make the task of assigning EBV latency state any easier (Brooks et al. 2009; Lee and Sugden 2008; Messinger et al. 2019). However, cellular gene expression associated with latency IIb versus low expression of LMP1 latency III indicates that it is feasible to dissect these states. Recent work by Amy Chadburn at Cornell (EBV Meeting 2022) further supports the clinical relevance of this state in the context of AIDS-associated lymphomas and post-transplant lymphomas. Nevertheless, we agree with the reviewer that it is confusing and difficult to follow, especially in this setting where the cell lines and xenografts don't seem to conform to strict latency types. As noted above, it may be best to simply accept the single cell heterogeneity of the system and characterize to the best of our ability using approaches such as IF/IHC and scRNAseq going forward. We have included this concern as a discussion point.

Comment R2.6: The variations in EBV gene expression based on the provided data is based on sequencing that will have differences based on depth and the percentage of cells infected with the virus in tumors. Saying there are differences without establishing a baseline to normalize expression is incorrect and at least should be followed up by validation of all the transcripts using RT-qPCR as well as IHC quantitation.(EBNA-3A, EBNA-3C

IHC mouse tumors) martin-rowe (EBNA-3 antibody)

Response: This point might be moot since we removed Figure 5E showing EBV gene expression from our RNAseq data that was not enriched for viral transcripts. However, we believe a sequencing depth of about 10 million reads per sample is sufficient at detecting EBV gene expression, and sequencing deeper than this would not have made any changes in the genes captured or expression profile. We were not comparing differences for transcripts expressed less than 1 in a million, and used the TMM (trimmed mean of M-values) method for normalization. The gene expression differences identified in this study were statistically tested and the differential expression analysis does take sequencing depth into account since the expression profiles are.

As stated in Comment R1.6, we added antibodies against EBNA3A and 3C to our IHC profiling (supplemental figure S6) and Qu-Path to quantify viral protein expression in our NSG-BL mouse tumors (Figure 5E).

Comment R2.7: The establishment of the cultures before implantation is a potential problem as within the window of 7-10 days or more cell duplication can introduce a new mutational burden that is distinct from the parental tumor. Why not implant the tumor directly without expansion to reduce this possibility as it makes interpretation of the data more complicated and potential for mistakes. It was also not clear how many total days of expansion were before implantation?

Response: Since our patients are in Kenya and the mice are in the USA, it is not feasible to transport tumors in a way that would maintain their viability for implantation. Although in the future we agree this would be worth pursuing once feasible to better validate generated lines or to examine in their own right. The advantage of our methods is that by establishing new BL cell lines, we generate a resource that can be shared, manipulated (to evict EBV for example) and used in a variety of model systems to learn more about BL and the role of EBV in BL pathogenesis. We show in Figure 3 that there are many characteristics retained by the tumor cell lines (i.e. host mutations, host gene expression) that are of interest and will be important to study further. We have clarified in the methods that the tumor cell lines are typically expanded for 1 week.

Comment R2.8: There is a mistake in figure 2 reference in the text for CD10 which is CD20 in Figure 2C.

Response: We apologize for this oversight. This typo has been corrected.

Comment R2.9: EBER signals from ISH is fuzzy in some of the pictures and the use of droplet PCR for copy number is difficult to reconcile as the standard curve was not determined or established based on the methods described. Missing negative control 40x as well as no specific probe control for EBER staining.

Response: We have clarified in the methods that the digital droplet PCR (ddPCR) does not need a standard curve since the technique gives absolute nucleic acid quantification based on positive droplet count numbers (Vogelstein and Kinzler 1999). Therefore its use to determine cell associated EBV viral load is appropriate and has been used previously in Forconi C. et al 2020. In our ddPCR experiments we included Namalwa and BL41 DNA as our positive and negative controls, respectively for EBV viral load. Namalwa and Jijoye cell line DNA was also used as EBV type 1 and type 2 controls respectively. We have now included the 40x negative control for the EBER staining in Figure 2E.

Comment R2.10: How was the 2 types of EBV determined by droplet PCR as both EBNA2 and EBNA3C will have varying sizes and its not clear how that would be identified using this Methodology.

Response: We designed type specific primers (not size based) and probes targeting these two EBV type specific genes. The ddPCR EBV typing was performed as previously described in Forconi C. et al 2020. The EBV typing protocol was a duplex reaction for the EBNA2, with a set of primers and probes targeting EBV type 1 and another set of primers and probes targeting EBV type 2 EBNA2. The EBNA3C typing reaction on the other hand had a set primer designed to amplify a region in the EBNA3C gene for both EBV type 1 and 2, while using 2 distinct probes targeting EBV type 1- EBNA3C and EBV type 2- EBNA3C gene. This method has been explained in the method section of this manuscript.

We also state in the revised manuscript: “We found minimal differences in EBV gene expression between type 1 and type 2 tumors, consistent with our previous publication using a larger sample size (Kaymaz et al. 2017)”.

Comment R2.11: Some of the analyses and the description of the results seems discordant to what is shown in the figures themselves as if the authors have not studied the data properly. The identification of some of the genes in figure 3. The PCA and volcano plots shows discrepancies between old and new BL cell lines but are stated as similar in text.

Response: The PCA and Volcano plots are two different results. The PCA in figure 3A compares old to new BL cell lines and eBL patient tumors showing that there are differences between them, while the volcano plot in the old Figure 3B was showing that they still share key genes used to define BL. The new Figure 3B shows the significantly enriched genes sets (in red and blue) that drive the genes differentially expressed between the old and the new BL cell lines, whereas the shared genes (gray dots) demonstrate that characteristic BL gene signatures are shared between the old and new cell lines. The point of these 2 figures is to demonstrate that important pathways may be differentially expressed in old compared to new BL cell lines which may influence interpretations of experimental studies involving these pathways. The data represented in the volcano plots are in Table S4 in the revised manuscript. We hope this clarifies your concern.

Comment R2.13: How are these genes STK33, MEK, ATF2 and HoxA9 differ in previous reports of BL sequencing in the data bases. Is this a unique finding? Maybe a heat map would be advantageous.

Response: Previous BL gene expression comparisons have been focused on comparing the tumor's genes expression profiles to that of other B cell lymphomas or the germinal center B cell as a control group. In this study, these findings regarding STK33 and ATF2 are novel and resulted from comparing the gene expression profile of previously established BL cell lines and the newly established BL cell lines. These findings are unique and are presented for the first time in this manuscript.

Comment R2.14: Tumor take of 2/6 if low not moderate and represents implantation and not tumour burden.

Response: We revised our sentence to read as follows: “BL719 avatars showed the poorest survival but with low tumor weights and a low implantation rate (2/6 mice) suggesting other pathologic mechanisms involved (Fig 5B)”

Comment R2.15: Stating that LMP1 was only expressed in one BL tumor is a bit of a reach as it should be validated for sensitivity and sequencing depth concerns that can arise with the RNASeq.

Response: We agree and have modified our statements regarding EBV latency patterns in the revised manuscript, based them on IHC protein expression patterns, and are explained in detail to responses to Reviewer 1 above.

Comment R2.16: The uniformity of the IHC staining in counterstain should be balanced and controls provided. Why did they use Daudi here as a control of Figure 5 instead of a well know BL cell line.

Response: We chose Daudi since this BL line had been previously used in this mouse model (Verma et al. 2017; Wang et al. 2018). But we have now included Akata and Mutu as additional, more classical BL cell lines, as suggested.

Comment R2.17: Signals for EBNA1 is lower than EBNA2 for 5E which is worrisome as EBNA1 should be most prominent signal for BL cells/tumors.

Response: The relative levels of staining are not directly comparable due to variation in primary antibody and secondary antibody. IHC is best used as a measure of on or off in the cell and that is as far as we interpret our results. We have now quantified the staining using Qu-Path which helps overcome these differences by comparing to negative background staining.

Comment R2.18: Some of the supplementary figures are unreadable and so not useful. Difficult to understand what the authors were trying to show when one cannot read the figures.

Response: Thank you for this feedback. We have carefully reviewed the supplementary figures and material making numerous changes to improve clarity and explain the relevance of each figure.

Comment R2.19 Overall: This is a potentially important study but the manuscript lacks careful analysis and experimental design, which makes interpretation of the data problematic. At this point there is not much more information that this manuscript provides that is not already known in the literature and at times complicates the field.

Response: We disagree that this manuscript does not contribute new and important findings or advance the field of BL. This is the first study to demonstrate that old BL cell lines have migrated too far away from BL tumors in patients, and thus continuing to use old BL cell lines will likely not be optimally informative to further our understanding of BL pathogenesis or move the field forward to discover better treatments for eBL patients. Not to mention that no one has made new eBL cell lines from African patients for over 30 years. Second, we clearly demonstrate that there is significant heterogeneity between eBL patient tumors. This heterogeneity is not only in terms of gene and protein expression but also led to differing outcomes in mouse models. That patient-derived BL tumors are heterogeneous both biologically and clinically has important implications for discovering new targeted treatments and customizing treatment regimens more likely to cure these children. This paper marks the beginning of a new era of equity and inclusion in research studies for children in Africa diagnosed with this cancer that are designed to find new and perhaps unique treatments. And, hopefully will inspire others to do similar cutting edge cancer studies in collaboration with low-income, limited resource countries.

January 6, 2023

RE: Life Science Alliance Manuscript #LSA-2021-01355-TR

Prof. Ann M. Moormann
University of Massachusetts Medical School
Department of Medicine
55 N Lake Ave
Worcester, MA 01655

Dear Dr. Moormann,

Thank you for submitting your revised manuscript entitled "Endemic Burkitt lymphoma avatar mouse models to explore tumor variation and treatment responses". We would be happy to publish your paper in Life Science Alliance pending final revisions necessary to meet our formatting guidelines.

- please address Reviewer 1's remaining comments
- please upload both your main and your supplementary figures as single, separate files and add a separate Figure Legend section to your main manuscript, including the legends for the main and supplementary figures, as well as the table legends
- please make sure your table files are uploaded as separate editable doc or excel files or that they are included in the doc file of your main manuscript text
- please add a Category for your manuscript to our system
- please add the Twitter handle of your host institute/organization as well as your own or/and one of the authors in our system
- please add the Author Contributions and a conflict of interest statement to the main manuscript text
- please use the [10 author names, et al.] format in your references (i.e. limit the author names to the first 10)
- please add a figure callout for Figure 5D and Figure S7 to your main manuscript text
- please double-check your figure callouts for Figure 3; you have a callout for Figure 3E, but this is not in the figure legend or in the figure

A. FINAL FILES:

B. MANUSCRIPT ORGANIZATION AND FORMATTING:

Sincerely,

Reviewer #1 (Comments to the Authors (Required)):

The revised manuscript is significantly improved and most of my concerns have been addressed. It will be a nice contribution to the field. I only have the following remaining concerns, which all pertain to the IHC and their interpretation:

-The text states that BL720 has the lowest % EBNA1 expression. From the images shown, it does not appear that BL720 has the lowest % EBNA1. Rather, BL719 appears to have the lowest EBNA1. I would suggest showing a bar graph with quantitation of EBNA1+ cells quantitated from a given number of cells with p-value, or perhaps just not making the claim.

-It is surprising that Mutu have only 24% EBNA1+ cells -- most cells should express EBNA1 if they have not lost EBV. An EBER stain could resolve this - perhaps this culture of Mutu has a large % of EBV- cells. Given the low rate of detection of EBNA1 in Mutu, I would be hesitant to make claims about the latency programs of the tumors based on similar immunostains, such as the claim that BL740 are in latency IIa/III. For instance, the stains for Mutu show EBNA2 in a subset of cells, but Mutu are not thought to be latency IIa. Also, according to the Price/Luftig reference cited, latency IIb should express not only EBNA2, but also EBNA-LP, EBNA3s and EBNA1, and yet very little EBNA3 expression was detected. I would hesitate to call BL740 latency IIb based on this definition, as only EBNA2 appears to be expressed. My opinion is that claims about the latency state are not supported by the data presented and should be toned down.

-It is written that LMP1 is only expressed in BL740, yet it is also expressed in some BL720. Would restate.

In response to the final comments from reviewer #1 we have made the following changes to our manuscript. **Our responses are bolded** below and highlighted in yellow in the revised version 2 of our manuscript.

Reviewer #1 (Comments to the Authors (Required)):

The revised manuscript is significantly improved and most of my concerns have been addressed. It will be a nice contribution to the field. I only have the following remaining concerns, which all pertain to the IHC and their interpretation:

-The text states that BL720 has the lowest % EBNA1 expression. From the images shown, it does not appear that BL720 has the lowest % EBNA1. Rather, BL719 appears to have the lowest EBNA1. I would suggest showing a bar graph with quantitation of EBNA1+ cells quantitated from a given number of cells with p-value, or perhaps just not making the claim.

Response 1: The IHC images are meant to be representative figures, whereas the graph shown in Figure 5E used an unbiased program to count the frequency of positively stained cells in numerous fields. Given the small sample size and multiple comparisons, a statistical analysis was not appropriate. However, we are tempering the interpretation of our findings. Since EBNA1 expression can vary and become below the level of detection by IHC, the main point we are making here is that all of our NSG-BL tumors expressed EBNA1 to some degree.

We have revised the statement in the manuscript to read as follows:

“We detected EBNA-1 expression in all our NSG-BL tumors but with varying frequencies ranging from 60% (BL717) to 20% or less (BL719 and BL720).”

-It is surprising that Mutu have only 24% EBNA1+ cells -- most cells should express EBNA1 if they have not lost EBV. An EBER stain could resolve this - perhaps this culture of Mutu has a large % of EBV- cells. Given the low rate of detection of EBNA1 in Mutu, I would be hesitant to make claims about the latency programs of the tumors based on similar immunostains, such as the claim that BL740 are in latency IIa/III. For instance, the stains for Mutu show EBNA2 in a subset of cells, but Mutu are not thought to be latency IIa. Also, according to the Price/Luftig reference cited, latency IIb should express not only EBNA2, but also EBNA-LP, EBNA3s and EBNA1, and yet very little EBNA3 expression was detected. I would hesitate to call BL740 latency IIb based on this definition, as only EBNA2 appears to be expressed. My opinion is that claims about the latency state are not supported by the data presented and should be toned down.

Response 2: As suggested by previous reviewers' comments, we included Mutu experiments in our revised manuscript because it is a classical latency I BL cell line expressing only EBNA1. By EBER staining, 46% of Mutu

tumors in our NSG mice were positive, which is consistent with EBER positivity of our other BL cell lines (Figure 2E). However, it is possible that these assays were not sensitive enough to detect low level expression. We also considered IHC counts below 2% to be background noise based on using Mutu as a negative control for the other viral proteins. We appreciate the reviewer's comments regarding the caveat of categorizing tumors into EBV latency patterns and have toned down our claims so as not to distract from the main message of this manuscript, i.e. 4 out of 5 of our patient-derived BL tumors did not have a fixed latency I pattern. The significance of this finding is being explored in more depth in a subsequent manuscript.

We have toned down our claims as follows:

“Inclusion of EBNA-2 expression for BL717, BL720 and BL725 suggests a latency IIb (Price and Luftig 2015), however the expression of the other viral proteins for this classification are missing. Negligible (less than 2%) expression of other viral proteins was considered background noise as compared to Mutu which is known to only express EBNA1.”

-It is written that LMP1 is only expressed in BL740, yet it is also expressed in some BL720. Would restate.

Response 3: The LMP1 expression levels were either non--existent or negligible (less than 2%) in all but BL740, so this was interpreted as background noise. We have added this limitation to our manuscript and included the following sentence:

“We found over 50% of BL740 cells expressing LMP1, suggesting a latency IIa pattern (Price and Luftig 2015). Negligible (less than 2%) LMP1 expression for BL720 and BL725 was considered background noise as compared to Mutu which is not known to express LMP1.”

February 1, 2023

RE: Life Science Alliance Manuscript #LSA-2021-01355-TRR

Prof. Ann M. Moormann
University of Massachusetts Medical School
Department of Medicine
55 N Lake Ave
Worcester, MA 01655

Dear Dr. Moormann,

Thank you for submitting your Research Article entitled "Endemic Burkitt lymphoma avatar mouse models to explore tumor variation and treatment responses". It is a pleasure to let you know that your manuscript is now accepted for publication in Life Science Alliance. Congratulations on this interesting work.

DISTRIBUTION OF MATERIALS:

Again, congratulations on a very nice paper. I hope you found the review process to be constructive and are pleased with how the manuscript was handled editorially. We look forward to future exciting submissions from your lab.

Sincerely,
